

**Terrestrial Ecosystem Process Model Biome-BGCMuSo:**
**Summary of improvements and new modeling possibilities**
Dóra Hidy[1], Zoltán Barcza[2], Hrvoje Marjanović[3], Maša Zorana Ostrogović Sever[3], Laura
Dobor[2], Györgyi Gelybó[4], Nándor Fodor[5], Krisztina Pintér[1], Galina Churkina[6], Steven
Running[7], Peter Thornton[8], Gianni Bellocchi[9], László Haszpra[10,13], Ferenc Horváth[11], Andrew
Suyker[12], Zoltán Nagy[1]
[1] MTA-SZIE Plant Ecology Research Group, Szent István University, Páter K. u.1., H-2103 Gödöllő, Hungary
[2] Department of Meteorology, Eötvös Loránd University, Pázmány P. sétány 1/A., H-1117 Budapest, Hungary
[3] Croatian Forest Research Institute, Department for Forest Management and Forestry Economics, Cvjetno
naselje 41, 10450 Jastrebarsko, Croatia
[4] Institute for Soil Sciences and Agricultural Chemistry, Centre for Agricultural Research, Hungarian Academy
of Sciences, Herman O. út 15., H-1163 Budapest, Hungary
[5] Agricultural Institute, Centre for Agricultural Research, Hungarian Academy of Sciences, Brunszvik u. 2., H-
15 2462
Martonvásár, Hungary
[6] Institute for Advanced Sustainability Studies e.V., Berliner Strasse 130, D-14467 Potsdam, Germany
[7] Numerical Terradynamic Simulation Group, Department of Ecosystem and Conservation Sciences University
of Montana, Missoula, MT 59812 USA
[8] Climate Change Science Institute / Environmental Sciences Division, Oak Ridge National Laboratory, Oak
Ridge, TN 37831-6335, USA
[9] UREP, INRA, 63000 Clermont-Ferrand, France
[10] Hungarian Meteorological Service, H-1675 Budapest, P.O.Box 39, Hungary
[11] Institute of Ecology and Botany, Centre for Ecological Research, Hungarian Academy of Sciences, Alkotmány
u. 2-4., H-2163 Vácrátót, Hungary
[12] School of Natural Resources, University of Nebraska-Lincoln, 806 Hardin Hall, Lincoln, Nebraska 68588,
USA
[13] Geodetic and Geophysical Institute, MTA Research Centre for Astronomy and Earth Sciences, H-9400
Sopron, Csatkai Endre utca 6-8., Hungary
*Correspondence to*: Dóra Hidy (dori.hidy@gmail.com)





**Abstract**. The process-based biogeochemical model Biome-BGC was enhanced to improve its ability to
simulate carbon, nitrogen and water cycles of various terrestrial ecosystems under contrasting management
activities. Biome-BGC version 4.1.1 was used as base model. Improvements included addition of new modules
such as the multilayer soil module, implementation of processes related to soil moisture and nitrogen balance,
soil moisture related plant senescence, and phenological development. Vegetation management modules with
annually varying options were also implemented to simulate management practices of grasslands (mowing,
grazing), croplands (ploughing, fertilizer application, planting, harvesting), and forests (thinning). New carbon
and nitrogen pools have been defined to simulate yield and soft stem development of herbaceous ecosystems.
The model version containing all developments is referred to as Biome-BGCMuSo (Biome-BGC with multi-
layer soil module). Case studies on a managed forest, cropland and grassland are presented to demonstrate the
effect of model developments on the simulation of plant growth as well as on carbon and water balance.
Keywords: model development, Biome-BGC, biogeochemical model, management, soil water content, drought
stress



## 1 Introduction

The development of climate models has led to the construction of Earth System Models (ESMs) with varying degrees of complexity where the terrestrial carbon cycle is included as a dynamic submodel (Ciais et al., 2013). In ESMs, the atmospheric concentration of carbon dioxide ($CO_2$) is no longer prescribed by the emission scenarios but it is calculated dynamically as a function of anthropogenic $CO_2$ emission and the parallel ocean and land surface carbon uptake. Focusing specifically on land surface, the carbon balance of terrestrial vegetation can be quantified by state-of-the-art biogeochemical models and are integral parts of the ESMs.

At present, there is no consensus on the future trajectory of the terrestrial carbon sink (Fig. 6.24 in Ciais et al., 2013). Some ESMs predict saturation of the land carbon sink in the near future while others show that the uptake will keep track with the increasing $CO_2$ emission. This means a considerable uncertainty related to the climate-carbon cycle feedback (Friedlingstein et al., 2006; Friedlingstein and Prentice, 2010). The wide range of model data related to the land carbon sink means that the current biogeochemical models have inherent uncertainties which must be addressed.

Biogeochemical models need continuous development to include empirically discovered processes and mechanisms e.g. acclimation processes describing the dynamic responses of plants to the changing environmental conditions (Smith and Dukes, 2012), regulation of stomatal conductance under elevated $CO_2$ concentration (Franks et al., 2013), drought effect on vegetation functioning (van der Molen et al., 2011), soil moisture control on ecosystem functioning (Yi et al., 2010) and other processes. Appropriate description of human intervention is also essential to adequately quantify lateral carbon fluxes and net biome production (Chapin et al., 2006). Keeping track with the new measurement-based findings is a challenging task but necessary to improve our ability to simulate the terrestrial carbon cycle more accurately.

The process-based biogeochemical model Biome-BGC is the focus of this study. The model was developed by the Numerical Terradynamic Simulation Group (NTSG), at the University of Montana (http://www.ntsg.umt.edu/project/biome-bgc), and is widely used to simulate carbon (C), nitrogen (N) and water fluxes of different terrestrial ecosystems such as deciduous and evergreen forests, grasslands and shrublands (Running and Hunt, 1993; Thornton, 1998; Thornton et al., 2002; Churkina et al., 2009; Hidy et al., 2012).

Biome-BGC is one of the earliest biogeochemical models that include an explicit N cycle module. It is now clear that climate-carbon cycle interactions are affected by N availability and the $CO_2$ fertilization effect can be limited by the amount of N in ecosystems (Friedlingstein and Prentice, 2010; Ciais et al., 2013). Therefore, the explicit simulation of the N cycle is an essential part of these biogeochemical models (Thomas et al., 2013).

Several researchers used and modified Biome-BGC in the past. Without being exhaustive, here we review some major applications on forest ecosystems, grasslands, croplands, urban environment, and we also list studies that focused on the spatial application of the model on regional and global scales.

Vitousek et al. (1988) studied the interactions in forest ecosystems such as succession, allometry and input-output budgets using Biome-BGC. Nemani and Running (1989) tested the theoretical atmosphere-soil-leaf area hydrologic equilibrium of forests using satellite data and model simulation with Biome-BGC. Korol et al. (1996) tested the model against observed tree growth and simulated the 5-year growth increments of 177 Douglas-fir



trees growing in uneven-aged stands. Kimball et al. (1997) used Biome-BGC to simulate the hydrological cycle
of boreal forest stands. Thornton et al. (2002) developed the model to simulate forest fire, harvest and replanting.
Churkina et al. (2003) used Biome-BGC to simulate coniferous forest carbon cycle in Europe. Pietsch et al.
(2003) presented developments of Biome-BGC where water infiltration from groundwater and the effect of
seasonal flooding were taken into account in floodplain forest areas. Bond-Lamberty et al. (2007) developed
Biome-BGC-MV to enable simulation of forest species succession and competition between vegetation types.
Vetter et al. (2005) used Biome-BGC to simulate the effect of human intervention on coniferous forest carbon
balance. Schmid et al. (2006) assessed the accuracy of Biome-BGC to simulate forest carbon balance in Central
Europe. Tatarinov and Cienciala (2006) further improved the Biome-BGC facilitating management practices in
forest ecosystems (including thinning, felling, and species change). Merganičová et al. (2005) and Petritsch et al.
(2007) implemented forest management in the model including thinning and harvest. Bond-Lamberty et al.
(2007) implemented elevated groundwater effect on stomatal conductance and decomposition in Biome-BGC.
Chiesi et al. (2007) evaluated the applicability of Biome-BGC in drought-prone Mediterranean forests. Turner et
al. (2007) used Biome-BGC to estimate the carbon balance of a heterogeneous region in the western United
States. Ueyama et al. (2010) used the model to simulate larch forest biogeochemistry in East Asia. Maselli et al.
(2012) used the model to estimate olive fruit yield in Tuscany, Italy. Hlásny et al. (2014) used Biome-BGC to
simulate the climate change impacts on selected forest plots in Central Europe.
Di Vittorio et al. (2010) developed Agro-BGC to simulate the functioning of C4 perennial grasses including a
new disturbance handler and a novel enzyme-driven C4 photosynthesis module.
Wang et al. (2005) applied Biome-BGC to simulate cropland carbon balance in China. Ma et al. (2011)
developed ANTHRO-BGC to simulate the biogeochemical cycles of winter crops in Europe including an option
for harvest and considering allocation to yield.
Trusilova and Churkina (2008) applied Biome-BGC to estimate the carbon cycle of urban vegetated areas.
Lagergren et al. (2006) used Biome-BGC to estimate the carbon balance of the total forested area in Sweden. Mu
et al. (2008) used Biome-BGC to estimate the carbon balance of China. Jochheim et al. (2009) presented forest
carbon budget estimates for Germany based on a modified version of Biome-BGC. Barcza et al. (2009, 2011)
used Biome-BGC to estimate the carbon balance of Hungary. Eastaugh et al. (2011) used Biome-BGC to
estimate the impact of climate change on Norway spruce growth in Austria. Biome-BGC was used within the
CARBOEUROPE-IP project as well as in several studies exploiting the related European scale simulation results
(e.g. Jung et al. 2007a, b; Vetter et al., 2008; Schulze et al., 2009; Ťupek et al., 2010; Churkina et al., 2010).
Hunt et al. (1996) used Biome-BGC in a global scale simulation and compared the simulated net primary
production (NPP) with satellite based vegetation index. Hunt et al. (1996) also used an atmospheric transport
model coupled with Biome-BGC to simulate surface fluxes to estimate the distribution of $CO_2$ within the
atmosphere. Churkina et al. (1999) used Biome-BGC in a global scale multimodel intercomparison to study the
effect of water limitation on NPP. Churkina et al. (2009) used Biome-BGC in a coupled simulation to estimate
the global carbon balance for the present and up to 2030. Biome-BGC is used in the Multiscale synthesis and
Terrestrial Model Intercomparison Project (MsTMIP) (Huntzinger et al., 2013; Schwalm et al., 2010) as part of
the North American Carbon Program.



In spite of its popularity and proven applicability, the model development temporarily stopped at version 4.2.
One major drawback of the model was its relatively poor performance in the simulation of the effect of
management practices. It is also known that anthropogenic effects exert a major role in the transformation of the
land surface in large spatial scales (Vitousek et al., 1998). Some structural problems also emerged like the
simplistic soil moisture module (using one soil layer), lack of new structural developments (see e.g. Smith and
Dukes, 2012), problems associated with phenology (Hidy et al., 2012), and lack of realistic response of
ecosystems to drought (e.g. senescence).
Our aim was to improve the model by targeting significant structural development, and to create a unified, state-
of-the-art version of Biome-BGC that can be potentially used in Earth System Models, as well as in situations
where management and water availability plays an important role (e.g. in semi-arid regions) in biomass
production. The success and widespread application of Biome-BGC can partly be attributed to the open source
nature of the model code. Following this tradition, we keep the developed Biome-BGC open source. In order to
support application of the model, a comprehensive User's Guide was also compiled (Hidy et al., 2015).
In the present work, the scientific basis of the developments is presented in detail followed by verification
studies in different ecosystems (forest, grassland and cropland) to demonstrate the effect of the developments on
the simulated fluxes and pools.
**2 Study sites**
Model developments were motivated by the need to improve model simulation for grasslands, croplands, and
forests. Here we demonstrate the applicability of the developed model at three sites on three different plant
functional types characterized by contrasting management, climate and site conditions.
**2.1 Grassland/pasture**
The large pasture near Bugacpuszta (46.69˚ N, 19.60˚ E, 111 m a.s.l.) is situated in the Hungarian Great Plain
covering an area of 1074 hectares. The study site (2 ha) has been used as pasture for the last 150 years according
to archive army maps from the 19$^{th}$ century. The soil surface is characterized by an undulating micro-topography
formed by winds within an elevation range of 2 m. The vegetation is highly diverse (species number over 80)
dominated by *Festuca pseudovina* Hack. ex Wiesb., *Carex stenophylla* Wahlbg., *Cynodon dactylon* L. Pers., *Poa*
spp.
The average annual precipitation is 562 mm and the annual mean temperature is 10.4 °C. According to the FAO
classification (Driessen et al., 2001) the soil type is chernozem with a rather high organic carbon content (51.5 g
kg$^{-1}$ for the 0-10 cm top soil; Balogh et al., 2011). The soil texture is sandy loamy sand (sand content 78%, silt
content 9% in 0-10 cm soil layer). The pasture belongs to the Kiskunság National Park and has been under
extensive grazing by a Hungarian grey cattle herd in the last 20 years. Stocking density was
0.23-0.58 animal ha$^{-1}$ during the 220-day-long grazing period between 2004 and 2012.
Carbon dioxide ($CO_2$) and latent heat flux (LHF) have been measured by the eddy-covariance (EC) technique.
Continuous measurements began in 2003 and they are used in this study as input and verification data (Nagy et
al., 2007, 2011). The EC station has a measurement height of 4 m and is equipped with a CSAT3 (Campbell





Scientific) sonic anemometer and a LI-7500 open path infrared gas analyzer (IRGA; LI-COR Inc., Lincoln, NE)
to measure fluxes of sensible and latent heat and $CO_2$. Several other sensors measure micrometeorological
variables including wind speed and direction, air temperature and relative humidity, precipitation, global,
reflected and net solar radiation, photosynthetically-active photon flux density (PPFD), reflected PPFD, soil heat
flux, soil temperature, and soil moisture. Soil water content was measured by CS616 (Campbell Scientific,
Shepshed, Leic's, UK) probes. Two probes were inserted horizontally at 3 and 30 cm depth and one was inserted
vertically averaging the soil water content of the upper 30 cm soil layer. For model evaluation measured soil
water content at 30 cm depth was used.
Data processing includes spike detection and removal following Vickers and Mahrt (1997) and linear detrending
to calculate fluctuations from the raw data. The flow disturbance effect of sensor heads is influenced by the angle
between the wind vector and horizontal plane, the so called angle of attack. To avoid errors caused by this error
our database is calibrated after van der Molen et al. (2004). The error caused by the inaccurate levelling of the
sonic anemometer is corrected by the planar fit method (Wilczak et al., 2001), in a slightly modified way.
Crosswind correction for sensible heat flux is done following Liu et al. (2001). The fluctuation in air density
influences the concentrations measured by the open path IRGA, which is not taken into account during the
measurements, but has to be done afterwards by the Webb-Pearman-Leuning correction (WPL; Webb et al.,
1980). To take into consideration the damping effect of sensor line averaging, separation distance between scalar
and wind sensor and the limited response time of both the anemometer and the LI-COR 7500 LI-COR Inc.,
Lincoln, NE, frequency response corrections were applied (Moore, 1986). Gap-filling and flux partitioning
method is based on the non-linear function between PPFD and daytime $CO_2$ fluxes, and temperature and night-
time $CO_2$ fluxes (Reichstein et al., 2005).

## 2.2  C4 cropland

Three production-scale cropland measurement sites were established in 2001 at the University of Nebraska
Agricultural Research and Development Center near Mead, Nebraska, USA, which are the part of the AmeriFlux
(http://ameriflux.ornl.gov/) and the FLUXNET global network (http://fluxnet.ornl.gov/). Mead1 (41.17° N,
96.48° W, 361 m a.s.l.; 48.7 ha) and Mead 2 (41.16° N, 96.47° W, 362 m a.s.l.; 52.4 ha) are both equipped with
center pivot irrigation systems. Mead3 (41.18° N, 96.44° W, 362 m a.s.l.; 65.4 ha) relies on rainfall. Maize is
planted each year at Mead1 while Mead2 and Mead3 are in a maize-soybean rotation.
On the Mead sites the annual average temperature is 10.1 °C and the mean annual precipitation total is 790 mm.
Soil at the site is deep silty clay loam consisting of four soil series: Yutan (fine-silty, mixed, superactive, mesic
Mollic Hapludalfs), Tomek (fine, smectitic, mesic Pachic Argialbolls), Filbert (fine, smectitic, mesic Vertic
Argialbolls), Filmore (fine, smectitic, mesic Vertic Argialbolls).
The $CO_2$ and energy fluxes are measured by an EC system using an omnidirectional three dimensional sonic
anemometer (Model R3, Gill Instruments Ltd., Lymington, UK), an open-path infrared $CO_2$/$H_2O$ gas analyzer
(Model LI-7500: LI-COR Inc., Lincoln, NE) and a closed-path $CO_2$/$H_2O$ system (Model LI-6262: LI-COR Inc.,
Lincoln, NE). The sensors were mounted 3 m above the ground when the canopy was shorter than 1 m, and later
moved to a height of 6 m until harvest when maize was planted (Suyker et al., 2004; Verma et al., 2005).





Raw EC data processing included correction of fluxes for inadequate sensor frequency response (i.e., tube
attenuation, sensor separation; Massman, 1991; Moore, 1986). Fluxes were adjusted for flow distortion (Nakai et
al., 2006) and the variation in air density due to the transfer of water vapour (Webb et al., 1980; Suyker et al.

4   2003).

Air temperature and humidity were measured at 3.0 m and 6.0 m height (Humitter50Y, Vaisala, Helsinki,
Finland). PPFD (LI 190SA Quantum Sensor, LI-COR Inc., Lincoln, NE), net radiation at 5.5 m height (Q*7.1,
Radiation and Energy Balance Systems Inc., Seattle, WA), and soil heat flux (0.06 m depth; Radiation and
Energy Balance Systems Inc.) were also measured (Verma et al., 2005). Soil water content measured at 0.1 m
depth was used in this study (ML2 Thetaprobe, Delta T Devices Ltd, Cambridge, UK).
Green leaf area index (LAI) was determined from destructive sampling. A LI-COR 3100 (LI-COR Inc., Lincoln,
NE) leaf area meter was used to measure sampled leaves. Each sampling was from six, 1-meter long row
samples from six parts of the field. Aboveground biomass was determined using destructive plant sampling.
Each sampling was from 1-meter long row samples from each of six different parts of the field. Plants were cut
off at ground level, brace roots were removed, and plants were placed in fine mesh bags. Plants were dried to
constant weight. Subsequently, the mass of green leaves, stems, reproductive organs, and senesced tissue were
determined on a per plant basis. Measured plant populations were used to convert from per plant-basis to a unit
ground area basis.
The Mead1 site was under no-till management prior to the harvest of 2005. Currently, there is a fall conservation
tillage for which approximately 1/3 of the crop residue is left on the surface. From 2010 to 2013, for a biomass
removal study, management at Mead2 was identical to Mead1 (continuous maize, fall conservation tillage, etc).
Management settings of the simulations were based on site records. In this study, Mead1 site is used for model
evaluation.

### 2.3 Deciduous broad-leaved forest

The Jastrebarsko site (45.62 °N 15.69 °E, 115 m a.s.l.) is a forest study site situated in a lowland oak forest that
is part of the state-owned Pokupsko basin forest complex, located approximately 35 km SW of Zagreb (Croatia).
Forest compartment where the EC tower for $CO_2$ flux measurement is located was a 37 years old mixed stand
dominated by *Quercus robur* L. (58 %) accompanied by other tree species, namely *Alnus glutinosa* (L.) Geartn.
(19%), *Carpinus betulus* L. (14%), *Fraxinus angustifolia* Vahl. (8%), and other (1%). *Corylus avellana* L. and
*Crateagus monogyna* Jacq. are common in the understory. Oak forests in this area are managed in 140 year long
rotations. Stands are thinned once every 10 years ending with regeneration cuts during the last 10 years of the
rotation (two or three), aimed at facilitating natural regeneration of the stand and continuous cover of the soil.
Annual mean temperature was 10.6 °C and average annual precipitation was 962 mm during the period of 1981–
2010 (data from the National Meteorological and Hydrological Service for the Jastrebarsko meteorological
station). The average annual depth to the groundwater ranges from 60 to 200 cm (Mayer, 1996). Groundwater
level in a 4 m deep piezometer was measured weekly during the vegetation season from 2008 onwards. During
winter and early spring, parts of the forest are waterlogged or flooded with stagnating water due to heavy soil.
Soil is mainly gleysol with low vertical water conductivity (Mayer, 1996), and according to the WRB, it is





classified as luvic stagnosol. The soil texture is dominantly clay with 18% of sand and 28% of silt fraction in the
0-30 cm top soil layer (Mayer, 1996).
Carbon dioxide and latent heat flux have been measured by the EC technique since September 2007 (Marjanović
et al., 2011a, b). The measurement height at the time of installation was 23 m above ground (3–5 m above the
top of the canopy). Since the forest stand grew, the measurement height was elevated to 27 m in April, 2011. The
EC system was made up of a sonic anemometer (81,000 V, R.M. Young, USA) and an open path IRGA (LI-
7500, LI-COR Inc., Lincoln, NE) with a sampling rate of 20 Hz. Meteorological measurements included soil
temperature, incoming short-wave radiation, incoming and outgoing PPFD, net radiation, air temperature and
humidity, soil heat flux and total rainfall. Soil water content was measured at 0–30 cm depth using two time-
domain reflectometers (Marjanović et al., 2011a).
Raw data processing of EC data was made using EdiRe Data Software (University of Edinburgh, United
Kingdom) according to the methodology based on the EuroFlux protocol (Aubinet et al., 2000) with a further
adjustment such as the WPL correction (Webb et al., 1980). Quality assessment and quality check analysis (QA
and QC) have been applied according to Foken and Vichura (1996). Net ecosystem exchange (NEE) was
obtained after taking into account the storage term which was calculated using a profile system designed to make
sequential $CO_2$ concentration measurements (IRGA, SBA-4, PPSystems) from 6 heights in a 6 minute cycle.
Quality assessment and quality control was made according to Foken and Vichura (1996). Gap-filling of missing
data and NEE flux partitioning to gross primary production (GPP) and total ecosystem respiration (TER) was
made using online EC data gap filling and flux partitioning tools (http://www.bgc-
jena.mpg.de/~MDIwork/eddyproc/) using the method of Reichstein et al. (2005).
Jastrebarsko forest stand characteristics data include diameter at breast height (dbh) and volume distribution by
tree species, weekly tree stem and height increment, tree mortality, phenology data, annual litterfall, litterfall C
and N content, fine root biomass, net annual root-derived carbon input, soil texture and carbon content
(Marjanović et al., 2011b; Ostrogović, 2013; Alberti et al., 2014). Data were used either as input in modeling
(e.g. groundwater level), for assessing some of the model parameters (e.g. phenology data) or model evaluation
(e.g. EC fluxes, tree increment measurements).

## 3 Description of base model: Biome-BGC 4.1.1 MPI

Our model developments started before Biome-BGC 4.2 was published. The starting point was Biome-BGC
v4.1.1 with modifications described in Trusilova et al. (2009). We refer to this model as Biome-BGC v4.1.1 MPI
version (MPI refers to the Max Planck Institute in Jena, Germany) or original Biome-BGC. Biome-BGC v4.1.1
MPI version was developed from the Biome-BGC family of models (Thornton, 2000). Biome-BGC is the
extension and generalization of the Forest-BGC model to describe different vegetation types including C3 and
C4 grasslands (Running and Coughlan, 1988; Running and Gower, 1991; Running and Hunt, 1993; Thornton,
2000; White et al., 2000; Trusilova et al., 2009).
Biome-BGC uses a daily time step and is driven by daily values of maximum and minimum temperatures,
precipitation, solar radiation, and daylight vapour pressure deficit (VPD). In addition to meteorological data
Biome-BGC uses site specific data (e.g. soil texture, site elevation, latitude), and ecophysiological data (e.g.



maximum stomatal conductance, specific leaf area, C:N mass ratios in different plant compartments, allocation related parameters, etc.) to simulate the biogeochemical processes of the given biome. The main simulated processes are photosynthesis, evapotranspiration, allocation, litterfall, C, N and water dynamics in the litter and soil (Thornton, 2000).

The three most important blocks of the model are the phenological, the carbon flux, and the soil flux block. The phenological block calculates foliage development and therefore affects the accumulation of C and N in leaf, stem (if present), root and consequently the amount of litter. In the carbon flux block gross primary production of the biome is calculated using Farquhar's photosynthesis routine (Farquhar et al., 1980) and the enzyme kinetics model based on Woodrow and Berry (1988). Autotrophic respiration is separated into maintenance and growth respirations. In addition to temperature, maintenance respiration is the function of the N content of living plant biomass, while growth respiration is calculated proportionally to the carbon allocated to the different plant compartments. The soil block describes the decomposition of dead plant material (litter) and soil organic matter, N mineralization and N balance in general (Running and Gower, 1991). The soil block uses the so-called converging cascade model (Thornton and Rosenbloom, 2005).

In Biome-BGC, the main parts of the ecosystem are defined as plant, soil and litter. Since Biome-BGC simulates water, carbon and N cycles of C3 and C4 plants, the following main pools are defined: leaf (C, N and water), fine root (C, N), soil (C, N and water) and litter (C, N). C and N pools have sub-pools (i.e., actual pools, storage pools and transfer pools). Sub-pools contain the amount of C or N available on the simulation day. Storage sub-pools store the amount that will appear next year (like a core or bud), while the transfer sub-pools store the whole content of the storage pool after the end of the actual transfer period until the next one, and what will be transferred gradually into the leaf carbon pool (like a germ) in the next transfer period.

The model simulation has two phases. The first is the spinup simulation (or in other words self-initialization or equilibrium run), which starts with very low initial level of soil C and N and runs until a steady state is reached with the climate in order to estimate the initial values of the state variables (Thornton, 2000; Thornton and Rosenbloom, 2005). In the second phase, the normal simulation uses the results of the spinup simulation as initial values for the C and N pools. This simulation is performed for a given, predetermined time period.

## 4 Methodological results: model adjustments

Some improvements focusing primarily on the development of the model to simulate carbon and water balance of managed herbaceous ecosystems have already been published (Hidy et al., 2012). Previous model developments included structural improvements on soil hydrology and plant phenology. Additionally, several management modules were implemented in order to provide more realistic fluxes for managed grasslands. We refer this model version as modified Biome-BGC (this is the predecessor of the current model).

Several developments have been made since the publication of the Hidy et al. (2012) study. The model that we present in this study is referred to as Biome-BGCMuSo (abbreviated as BBGCMuSo, where MuSo refers to MUltilayer SOil module). Currently, BBGCMuSo has several versions but hereafter only the latest version (4.0) is discussed.





In this paper, we provide detailed documentation of all changes made in the model logic compared to the original
model (Biome-BGC v4.1.1). Here we briefly mention the previously published developments and we provide
detailed description concerning the new features. In order to illustrate the effect and the importance of the
developments, model evaluations are presented for three contrasting sites equipped with EC measurement
systems in Section 5.
Due to the modifications of the model structure and the implementation of the new management modules, it was
necessary to modify the structure of the input and output files of the model. Technical details are discussed in the
User's        Guide        of        BBGCMuSo        (Hidy        et        al.,        2015;
http://nimbus.elte.hu/bbgc/files/Manual_BBGC_MuSo_v4.0.pdf).
We summarized the model adjustments in Table 1. Within the table the implemented processes are grouped for
clarity. Below we document the adjustments in detail.

### 4.1 Multilayer soil

The predecessor of Biome-BGC (FOREST-BGC) was developed to simulate the carbon and water budgets of
forests where soil moisture limitation is probably less important due to the deeper rooting zone. Therefore, its
soil sub-model was simple and included only a one-layer budget model. Due to the recognized importance of soil
hydrology on the carbon and water balance, state-of-the-art biogeochemical models include multilayer soil
module (e.g. Schwalm et al., 2015).
In order to improve the simulation quality of water and carbon fluxes with Biome-BGC, a seven-layer soil sub-
model was implemented. Seven layers provide an optimal compromise between simulation accuracy and
computational cost. In accordance with the soil layers, we also defined new fluxes within the model. Note that in
the Hidy et al. (2012) publication the developed Biome-BGC had only 4 soil layers, so the 7-layer module is an
improvement.
The first layer is located at depth of 0-10 cm; the second is at 10-30 cm, the third is at 30-60 cm, the fourth is at
60-100 cm, the fifth is at 100-200 cm, and the sixth is at 200-300 cm. The bottom (7[th], hydrologically and
thermally inactive) layer is located at depth 300-1000 cm. The depth of a given soil layer is represented by the
center of the given layer. Below 5 meters the soil temperature can be assumed equal to the annual average air
temperature of the site (Florides and Kalogirou, 2016). Furthermore, in the bottom layer we assume that soil
water content (SWC) is equal to the field capacity (constant value) and soil mineral content has a small, constant
value (1 kg N ha$^{-1}$ within the 7 m deep soil column). The percolated water (and soluble N) to the bottom,
hydrologically inactive layer is a net loss, while the water (and soluble N) diffused upward from the bottom layer
is a net gain for the simulated soil system.
Soil texture and soil bulk density can be defined by the user layer by layer. If the maximum rooting zone
(defined by user) is greater than 3 m, the roots reach the constant boundary soil layer and water uptake can occur
from that layer, as well. In previous BBGCMuSo model versions, the soil texture was constant with depth and
the maximum possible rooting depth was 5 m.





### 4.1.1 Soil thermodynamics

Since some of the soil processes depend on the actual soil temperature (e.g. decomposition of soil organic matter), it is necessary to calculate soil temperature for each active layer. Given the importance of soil temperature (e.g. Sándor et al., 2016), the soil module was also reconsidered in BBGCMuSo. The daily soil surface temperature is determined after (Zheng et al., 1993). The basic equations are detailed in Hidy et al. (2012). An important modification in BBGCMuSo is that the average temperature of the top soil layer (with thickness of 10 cm) is not equal to the above mentioned daily surface temperature. Instead, temperature of the active layers is estimated based on an empirical equation with daily time step.

Two optional empirical estimation methods are implemented in BBGCMuSo. The first is a simple method, assuming a logarithmic temperature gradient between the surface and the constant temperature boundary layer. This is an improvement compared to the modified Biome-BGC where a linear gradient was assumed. The second method is based on the soil temperature estimation method of the DSSAT model family (Ritchie 1998) and the 4M model (Sándor and Fodor, 2012).

### 4.1.2 Soil hydrology

The C and the hydrological cycles of the ecosystems are strongly coupled due to interactions between soil moisture and stomatal conductance, soil organic matter decomposition, N mineralization and other processes. Accurate estimation of the soil water balance is thus essential.

Among soil hydrological processes, the original Biome-BGC only takes into account plant uptake, canopy interception, snowmelt, outflow (drainage) and bare soil evaporation. We have added the simulation of runoff, diffusion, percolation, pond water formation and enhanced the simulation of the transpiration processes in order to improve the soil water balance simulation. Optional handling of seasonally changing groundwater depth (i.e., possible flooding due to elevated water table) is also implemented. In the Hidy et al. (2012) study, only runoff, diffusion and percolation were implemented (the simulation of these processes has been further developed since then). Pond water formation and simulation of groundwater movement are new features in BBGCMuSo 4.0.

Estimation of characteristic soil water content values

There are four significant characteristic points of the soil water retention curve: saturation, field capacity, permanent wilting point and hygroscopic water. Beside volumetric SWC, soil moisture status can also be described by soil water potential (PSI; MPa). The Clapp-Hornberger parameter (B; dimensionless) and the bulk density (BD; g cm$^{-3}$), are also important in soil hydrological calculations (Clapp and Hornberger, 1978). Soil texture information (fraction of sand and silt within the soil; clay fraction is calculated by the model internally as a residual) for each soil layers are input data for BBGCMuSo. Based on soil texture other soil properties (characteristic points of SWC, B, BD) can be estimated internally by the model using pedotransfer functions (Fodor and Rajkai, 2011). The default values are listed in Table 2, which can be adjusted by the user within their plausible ranges.

As in the original Biome-BGC, the PSI at saturation is the function of soil texture in BBGCMuSo:

$$PSI_{sat} = -\left\{ exp\left[ (1.54 - 0.0095 \cdot SAND + 0.0063 \cdot SILT) \cdot ln(10) \right] \cdot 9.8 \cdot 10^{-5} \right\} \tag{1}$$





where SAND and SILT are the sand and silt percent fractions of the soil, respectively. In BBGCMuSo, PSI for
unsaturated soils is calculated from SWC using the saturation value of SWC (SWCsat) and the B parameter:
$$\mathrm{PSI} = \exp\left(\frac{\mathrm{SWC_{sat}}}{\mathrm{SWC}} \cdot \ln(B)\right) \cdot \mathrm{PSI_{sat}},$$     (2)
Pond water and hygroscopic water
In case of intensive rainfall events, when not all of the precipitation can infiltrate, pond water is formulated at the
surface. Water from the pond can infiltrate into the soil after water content of the top soil layer decreases below
saturation. Evaporation of the pond water is assumed to be equal to potential soil evaporation.
SWC (as well as C and N content) is not allowed to become negative. The theoretical lower limit of SWC is the
hygroscopic water, i.e., the water content of air-dried soil. Therefore, in case of large calculated
evapotranspiration (calculated based on the Penman-Monteith equation as driven by meteorological data) and
dry soil, the soil water pool of the top layer can be depleted (approaching hygroscopic water content). In this
case evaporation and transpiration fluxes are limited in BBGCMuSo. Hygroscopic water content is also the
lower limit in decomposition calculations (see below).
*Runoff*
Our runoff simulation method is semi-empirical and uses the precipitation amount and the Soil Conservation
Service (SCS) runoff curve number (Williams, 1991). If the precipitation is greater than a critical amount which
depends on the water content of the topsoil, a fixed part of the precipitation is lost due to runoff and the rest
infiltrates into the soil. The runoff curve number can be set by the user or can be estimated by the model (Table

19    2).

Percolation and diffusion
In BBGCMuSo, two calculation methods of vertical soil water movement are implemented. The first method is
based on Richards' equation (Chen and Dudhia, 2001; Balsamo et al., 2009) Detailed description and equations
for this method can be found in Hidy et al. (2012). The second method is the so-called 'tipping bucket method'
(Ritchie, 1998) which is based on semi-empirical estimation of percolation and diffusion fluxes and has a long
tradition in crop modelling.
In case of the first method hydraulic conductivity and hydraulic diffusivity are used in diffusion and percolation
calculations. These variables change rapidly and significantly upon changes in SWC. Though the main processes
are calculated on daily time-step within the model, the simulation of the soil hydrological parameters required
finer temporal resolution; therefore a nested variable time-step-based calculation was introduced into the soil
hydrology sub-module in case of the first calculation method (in case of the tipping bucket method, a daily time-
step is used).
In BBGCMuSo, the time-step (TS; seconds) of the soil water module integration is dynamically changed. In
contrast to the modified Biome-BGC (Hidy et al., 2012) the TS is based on the theoretical maximum of soil
water flux instead of the amount of precipitation.
As a first step we calculate water flux data for a short, 1-second TS for all soil layers based on their actual SWC.
The most important problem here is the selection of the optimal TS: too small TS values result in slow model
run, while too large TS values lead to overestimation of water fluxes. TS is estimated based on the magnitude of





the maximal soil moisture content change in 1 second ($\Delta SWC_{max}$; $m^3$ $m^{-3}$ $s^{-1}$). Equation (3)-(4) describes the
calculation of TS:
$$TS = 10^{EXPON} \qquad . \tag{3}$$
where
$$\begin{aligned} &EXPON = abs\left[LV + (DL + 3)\right] \text{ if } DL + LV < 0 \\ &EXPON = 0 \qquad\qquad\qquad \text{ if } DL + LV \geq 0 \end{aligned} \tag{4}$$
LV is the rounded local value of the maximal soil moisture content change, and DL is the discretization level,
which can be 0, 1 or 2 (low, medium or high discretization level which can be set by the user).
Therefore, if the magnitude of $\Delta SWC_{max}$ is $10^{-3}$ $m^3$ $m^{-3}$ $s^{-1}$ (e.g. SWC increases from 0.400 $m^3$ $m^{-3}$ to
0.401 $m^3$ $m^{-3}$ in 1 second), then LV is -3, so TS is $10^0$ = 1 second (if DL=0). The 1-second TS is necessary if
$\Delta SWC_{max}$ is above $10^{-1}$ $m^3$ $m^{-3}$ $s^{-1}$. Using daily TS proved to be adequate when $\Delta SWC_{max}$ was below
$10^{-8}$ $m^3$ $m^{-3}$ $s^{-1}$.
After calculating the first TS (TSstep1 = 1 seconds), water fluxes are determined by using the pre-calculated
time steps, and the SWC of the layers are updated accordingly. In the next n steps the calculations are repeated
until reaching a threshold (86400, the number of the seconds in a day). This method helps to avoid the
overestimation of the water fluxes while also decreasing the computation time of the model.
Groundwater
Poorly drained forests (e.g. in boreal regions or in lowland areas) are special ecosystems where groundwater and
flooding play an important role in soil hydrology and plant growth (Bond-Lamberty et al., 2007; Pietsch et al.,
2003). In order to enable groundwater effects (vertically varying soil water saturation), we implemented an
option in BBGCMuSo to supply external information about the depth of the water table. Groundwater depth is
controlled by prescribing the depth of saturated zone (groundwater) within the soil. We note that the
groundwater implementations by Pietsch et al. (2003) and by Bond-Lamberty et al. (2007) are different from our
approach as the they calculate water table depth internally by the modified Biome-BGC. During the spinup phase
of the simulation, the model can only use daily average data for one typical simulation year (i.e., multi-annual
mean water table depth). During the normal phase, the model can read daily groundwater information defined
externally.
The handling of the externally supplied, near-surface groundwater information is done as follows. If the upward
moving water table reaches the bottom border of a simulated soil layer, part of the given layer becomes "quasi-
saturated" (99% of the saturation value) and thus the average soil moisture content of the given layer increases.
If the water table reaches the upper border of the given soil layer, then the given layer becomes quasi-saturated.
The quasi-saturation state allows for the downward flow of water through a saturation soil layer, particularly
through the last layer. This approach was necessary because of the discrete size of soil layers and the fact that the
input groundwater level data constitute a net groundwater level. Namely, groundwater level is partly affected by
lateral groundwater flows (which are unknown) and corresponding changes in hydraulic pressure that can push
the water up; as well it is partly affected by draining of the upper soil layers. While soil is draining it is possible
that the groundwater level remains unchanged, or that its level decreases slower because the drained water from
the upper layer replaces the water that leaves the system (i.e., drains to below 10 m, which is the bottom of the



last soil layer). With implementation of quasi-saturation state we allowed for the water to flow through saturated
layers, in particular through the bottom one. Otherwise the water could become "trapped" because the model
routine could not allow downward movement of water from an upper soil layer to a lower layer that is already
saturated (because that layer would have already been "full").
Groundwater level affects soil water content and in turn affects stomatal conductance and soil organic matter
decomposition (see below).
**4.1.3 Root distribution**
Water becomes available to the plant through water uptake by the roots. The maximum depth of the rooting zone
is a user-defined parameter in the model. For herbaceous vegetation, the temporally changing depth of the root is
simulated based on an empirical, sigmoid function (Campbell and Diaz, 1988). In forests, fine root growth is
assumed to occur in the entire root zone and rooting depth does not change with time. This latter logic is used
due to the presence of coarse roots in forests which are assumed to change depth slowly (in juvenile forests this
approach might be problematic).
In order to weight the relative importance of the soil layers (i.e., to distribute total transpiration or root
respiration among soil layers), it is necessary to calculate the distribution of roots in the soil layers. The
proportion of the total root mass in the given layer ($R_{layer}$) is calculated based on empirical exponential root
profile approximation after Jarvis (1989):

$$R_{layer} = f \cdot \left( \frac{\Delta z_{layer}}{z_r} \right) \cdot \exp\left[ -f \cdot \left( \frac{z_{layer}}{z_r} \right) \right] \quad . \tag{5}$$

In Eq. (5), f is an empirical root distribution parameter (its proposed value is 3.67 after Jarvis, 1989), $\Delta z_{layer}$ and
$z_{layer}$ is the thickness and the midpoint of the given soil layer, respectively, and $z_r$ is the actual rooting depth.
Rooting depth and root distribution are taken into account by all ecophysiological processes which are affected
by soil moisture content (transpiration), soil carbon content (maintenance respiration) and soil N content
(decomposition).
**4.1.4 Nitrogen budget**
In previous model versions, uniform distribution of mineral N was assumed within the soil profile. In
BBGCMuSo, we hypothesize that varying amounts of mineralized N are available within the different soil layers
and are available for root uptake and other losses. The change of soil mineral N content is calculated layer by
layer in each day. In the root zone (i.e., within soil layers containing roots), the changes of mineralized N-content
are caused by soil processes (decomposition, microbial immobilization, denitrification), plant uptake, leaching,
atmospheric deposition and biological N fixation. The produced/consumed mineralized N (calculated by
decomposition and daily allocation functions) is distributed within the layers depending on their soil mineral N
content. Mineralized N from atmospheric N deposition (Ndep) increases the N content of the first (0-10 cm) soil
layer. Biological N fixation is divided between root zone layers as a function of the root fraction in the given
layer. In soil layers without roots, N content is affected by transport (e.g. leaching) only. Leaching is calculated
based on empirical function using the proportion of soluble N that is subject to mobilization (an adjustable





ecophysiological parameter in the model), mineral N content and the water fluxes (percolation and diffusion) of
the different soil layers.
There is an additional mechanism for N loss within the soil. According to the model logic (Thornton and
Rosenbloom, 2005) if there is excess mineral nitrogen in the soil following microbial immobilization and plant
N uptake, it is subject to volatilization and denitrification as a constant proportion of the excess mineral nitrogen
pool for a given day. In the original Biome-BGC (v4.1.1) this proportion was defined by a fixed parameter
within the model code. In Biome-BGC v4.2 two parameters were introduced that control bulk denitrification:
one for the wet and one for dry case (the wet case is defined when SWC is greater than the 95% of saturation). In
BBGCMuSo we implemented the Biome-BGC v4.2 logic which means that these values can be adjusted as part
of the ecophysiological model parameterization
**4.1.5  Soil moisture stress index**
Stomatal closure occurs due to low relative atmospheric moisture content (high VPD), and insufficient soil
moisture (drought stress; Damour et al., 2010) but also due to anoxic conditions (e.g. presence of elevated
groundwater or high soil moisture content during and after large precipitation events; Bond-Lamberty et al.,
2007). The original Biome-BGC had a relatively simple soil moisture stress function which was not adjustable
and was unable to consider anoxic conditions.
In order to create a generalized model logic for BBGCMuSo, the soil moisture limitation calculation was
improved, and the use of the stress function on stomatal conductance (and consequently transpiration and
senescence) calculations was extended taking into account both types of limitations mentioned above (limit1:
drought, limit2: anoxic condition close to soil saturation). The start and end of the water stress period are
determined by comparing actual and predefined, characteristic points of SWC and calculating a novel soil
moisture stress index (SMSI; dimensionless). Characteristic points of soil water status can be defined using
relative SWC (relative SWC to field capacity in case of limit1 and to saturation in case of limit2) or soil water
potential (see Section 3.4 of Hidy et al., 2015).
SMSI is the function of normalized soil water content (NSWC) (in contrast to original model, where SMSI was
the function of soil water potential). NSWC is defined by the following equation:

$$\begin{aligned} NSWC &= \frac{SWC - SWC_{wp}}{SWC_{sat} - SWC_{wp}}, \text{if } SWC_{wp} < SWC \\ NSWC &= 0 \qquad\qquad\qquad , \text{if } SWC_{wp} \geq SWC \end{aligned} \qquad (6)$$

where SWC is the soil water content of the given soil layer ($m^3\ m^{-3}$), $SWC_{wp}$ and $SWC_{sat}$ are the wilting point
and the saturation value of soil water content, respectively. Parameters can either be set by the user or calculated
by the model internally (see Table 2).
The NSWC and the soil water potential as function of SWC are presented in Fig. 1 for three different soil types:
sand soil (SAND: 90%, SILT: 5%), sandy clay loam (SAND: 50%, SILT: 20%), and clay soil (SAND: 8%,
SILT: 45%).
According to our definition SMSI is a function of NSWC and can vary between 0 (maximum stress) and 1
(minimum stress).





The general form of SMSI is defined by the following equations:

$$SMSI = \frac{NSWC}{NSWC_{crit1}}, \quad \text{if } NSWC < NSWC_{crit1}$$

$$SMSI = 1, \qquad \qquad \text{if } NSWC_{crit1} < NSWC \leq NSWC_{crit2} \qquad (7)$$

$$SMSI = \frac{1 - NSWC}{1 - NSWC_{crit2}}, \text{if } NSWC_{crit2} < NSWC$$

where $NSWC_{crit1}$ and $NSWC_{crit2}$ ($NSWC_{crit1} < NSWC_{crit2}$) are the characteristic points of the normalized soil
water curve, calculated from the relative soil water content or soil water potential values defined by
ecophysiological parameters. Conversion from relative values to NSWC is made within the model. Characteristic
point $NSWC_{crit1}$ is used to control drought related limitation, while $NSWC_{crit2}$ is used to control excess water
related limitation (e.g. anoxic soil related stomatal closure).
The shape of the soil stress function in the original model (based on soil water potential) and in BBGCMuSo
(based on NSWC) are presented in Fig. 2.
The value of the SMSI is zero in case of full soil water stress (below the wilting point). It starts to increase at the
wilting point (which depends on soil type; Table 1) and reaches its maximum (1) at the SWC where water stress
ends. This latter characteristic value can be set by the user. In this example (Fig. 2), field capacity was used as
the characteristic value. The new feature of the BBGCMuSo is that beyond the optimal soil moisture content
range, the soil stress can decrease again (i.e., increasing stress) due to saturation. This second characteristic value
(limit2) can also be set by a model parameter. In the example of Fig. 2 95% of the saturation value was used as
the second characteristic value.
The model requires only a single soil moisture stress function to calculate stomatal conductance, while soil water
status is calculated layer by layer. Therefore, an average stress function for the total rootzone is necessary, which
is the average of the layer factors weighted by root fraction in each layer.
Beside stomatal conductance, SMSI is used in the transpiration calculation in the multilayer soil. Instead of the
averaged soil water status of the whole soil column (as in original Biome-BGC), in BBGCMuSo, the
transpiration flux is calculated layer by layer. The transpiration flux of the ecosystem is assumed to be equal to
the total root water uptake on a given day ($TRP_{sum}$). The transpiration calculation is based on the Penman-
Monteith equation using stomatal conductance (this feature is the same as in the original Biome-BGC method).
The transpiration fluxes are divided between layers ($TRP_{layer}$) according to the soil moisture limitation of the
given layer ($SMSI_{layer}$) and the root fraction in the given layer ($R_{layer}$ in Eq. 5):

$$TRP_{layer} = TRP_{sum} \cdot \frac{SMSI_{layer} \cdot R_{layer}}{SMSI_{sum}} \qquad (8)$$

where $SMSI_{sum}$ is the sum of the $SMSI_{layer}$ values in the rootzone.
According to the modifications, if the soil moisture limitation is full (SMSI=0), no transpiration can occur.
**4.1.6 Senescence calculation**
The original Biome-BGC ignores plant wilting and associated senescence (where the latter is an irreversible
process) caused by prolonged drought. In order to solve this problem, a new module was implemented to





simulate the ecophysiological effect of drought stress on plant mortality. A senescence simulation has already
been implemented in developed Biome-BGC (Hidy et al., 2012) and it was further improved in BBGCMuSo
following a different approach. In BBGCMuSo, if the plant available SWC decreases below a critical value, the
new module starts to calculate the number of the days under drought stress. Due to low SWC during a prolonged
drought period, aboveground and belowground plant material senescence is occurring (actual, transfer and
storage C and N pools) and the wilted biomass is translocated into the litter pool.
A so-called "soil stress effect" ($SSE_{total}$) is defined to calculate the amount of plant material that wilts due to cell
death in one day due to drought stress. The $SSE_{total}$ quantifies the severity of the drought for a given day.
Severity of drought is a function of the number of days since soil moisture stress is present (NDWS). Soil water
stress is assumed if the averaged SMSI of the root zone is less than a critical value (which is an adjustable
ecophysiological parameter in the model). It is assumed that after a longer time period with soil moisture stress,
the senescence is complete and no living non-woody plant material remains. This longer time period is
quantified by an ecophysiological parameter, which is the "critical number of stress days after which senescence
mortality is complete".
The $SSE_{total}$ varies between 0 (no stress) and 1 (total stress). According to the BBGCMuSo logic, the $SSE_{total}$ is
the function of SMSI, NDWS, $NDWS_{crit}$, and $SMSI_{crit}$:

$$SSE_{total} = SSE_{SMSI} \cdot SSE_{NDWS}$$
$$SSE_{SMSI} = \left(1 - \frac{SMSI}{SMSI_{crit}}\right) ; \text{if } SMSI < SMSI_{crit}$$
$$SSE_{SMSI} = 0 \qquad ; \text{if } SMSI \geq SMSI_{crit} \qquad\qquad (9)$$
$$SSE_{NDWS} = \frac{NDWS}{NDWS_{crit}} \quad ; \text{if } NDWS < NDWS_{crit}$$
$$SSE_{NDWS} = 1 \qquad ; \text{if } NDWS \geq NDWS_{crit}$$

The $SEE_{total}$ is used to calculate the actual value of non-woody aboveground and belowground mortality defined
by SMCA and SMCB, respectively. SMCA defines the fraction of living C and N that dies in one day within
leaves and herbaceous stems. SMCB does the same for fine roots. SMCA and SMCB are calculated using two
additional ecophysiological parameters: minimum mortality coefficient (minSMC) of non-woody aboveground
(A) and belowground (B) plant material senescence (minSMCA and minSMCB, respectively):

$$SMCA = minSMCA + (1 - minSMCA) \cdot SSE_{total}$$
$$SMCB = minSMCB + (1 - minSMCB) \cdot SSE_{total} \qquad\qquad (10)$$

Parameters minSMCA and minSMCB can vary between 0 and 1. In case of 0, no senescence occurs. In case of
1, all living C and N will die within one day after the occurrence of drought stress. In this sense SMCA and
SMCB can vary between their minimum value (minSMCA and minSMCB, respectively, if $SSE_{total}$ is zero) and 1
(if $SSE_{total}$ is 1). This latter case occurs when NDWS>$NDWS_{crit}$ and SMSI=1.
SMCA is used to calculate the amount of non-woody aboveground plant material (leaves and soft stem)
transferred to the standing dead biomass pool (STDB; kgC m$^{-2}$) due to soil moisture stress related mortality on a
given day. Note that STDB is a temporary pool from which C and N contents transfer to the litter pool gradually;
the concept of STDB is a novel feature in the model. In case of belowground mortality, it defines the amount of





fine root that goes to the litter pool directly. The actual senescence ratio is calculated as SMCA (and SMCB)
multiplied with the actual living C and N pool.
Fig. 3 demonstrates the senescence calculation with an example. The figure shows the connections between the
change of SWC, normalized soil water content, SMSI, different types of soil stress effects (SSE$_{SMSI}$, SSE$_{NDWS}$,
SSE$_{total}$) and senescence mortality coefficient as function of number of days since water stress is present. The
figure shows a theoretical situation in which SWC decreases from field capacity to hygroscopic water within 30
days. The calculation refers to a sandy soil (SWC$_{sat}$ = 0.44 m$^3$ m$^{-3}$, SWC$_{fc}$ = 0.25 m$^3$ m$^{-3}$, SWC$_{wp}$ = 0.09 m$^3$ m$^{-3}$;
SWC$_{fc}$ refers to SWC at field capacity). In this example, we assumed that that relSWC$_{crit1}$ is 1.0 (field capacity)
and relSWC$_{crit2}$ = 0.9 (10% below saturation), the critical number of stress days after which senescence mortality
is complete (NDWS$_{crit}$) is 30 and the critical soil moisture stress index (SMSI$_{crit}$) is 0.3.
**4.1.7 Decomposition and respiration processes**
Within Biome-BGC, decomposition processes of litter and soil organic matter are influenced by soil
temperature, soil water status, soil and litter C and N content, while root maintenance respiration is affected by
soil temperature and root C and N content. In BBGCMuSo, the soil moisture and temperature limitation effects
are calculated layer by layer based on the SWC and soil temperature of the given soil layer (instead of the
averaged soil water status or soil temperature of the whole soil column as in the original model). The C and N
contents are also calculated layer by layer from the total C and N content of the soil column weighted by the
proportion of the total root mass in the given layer.
The maintenance root respiration flux is calculated based on the following equation:
$$MR(\text{root}) = \sum_{\text{layer}=1}^{nr} \left( N_{\text{root}} \cdot R_{\text{layer}} \cdot \text{mrpern} \cdot Q_{10}^{\frac{T(\text{soil})_{\text{layer}}-20}{10}} \right) \qquad (11)$$
where n$_r$ is the number of the soil layers which contain root, N$_{root}$ is the total N content of the soil, R$_{layer}$ is the
proportion of the total root mass in the given layer, mrpern is an adjustable ecophysiological parameter
(maintenance respiration per kg of tissue N), Q$_{10}$ is the fractional change in respiration with a 10°C temperature
change and T(soil)$_{layer}$ is the soil temperature of the given layer.
There are eight types of non-N limited fluxes between litter and soil compartments. These fluxes are the function
of soil and litter C or N content, soil moisture and soil temperature stress functions. The most important
innovation is that total decomposition fluxes are calculated as the sum of partial fluxes regarding to the given
layer similarly to respiration flux. The soil temperature function is the same as in the original Biome-BGC. The
soil moisture stress function is a linear function of SWC in contrast to a logarithmic function of soil water
potential in the original Biome-BGC. A major development here is that beside drought effect, anoxic stress is
also taken into account because anoxic condition caused by saturation can affect decomposition of soil organic
matter (thus N mineralization; Bond-Lamberty et al., 2007). The shape of the modified stress index (SMSI$_{decomp}$)
is similar to the one presented in Bond-Lamberty et al. (2007).
BBGCMuSo uses the following stress index (with value between 0 and 1) to control decomposition in response
to changing SWC in a given layer:





$$SMSI_{decomp} = \frac{SWC - SWC_{hyg}}{SWC_{opt} - SWC_{hyg}}, \text{if } SWC \leq SWC_{opt}$$

$$SMSI_{decomp} = \frac{SWC_{sat} - SWC}{SWC_{sat} - SWC_{opt}}, \text{if } SWC_{opt} < SWC$$

(12)

where SWC, $SWC_{hyg}$ and $SWC_{sat}$ is the actual soil water content, hygroscopic water and the saturation values of
the given soil layer, respectively. $SWC_{opt}$ is calculated from the relative SWC for soil moisture limitation which
is a user supplied ecophysiological input parameter. The limitation function of decomposition in the original
Biome-BGC (based on PSI) and in BBGCMuSo (based on SWC) are presented in Fig. 4.
In case of the original model soil stress index starts to increase at -10 MPa (this value is fixed in the model) and
reaches its maximum at saturation. In the case of BBGCMuSo, soil stress index starts to increase at hygroscopic
water and reaches its maximum at optimal SWC (which is equal to the $NSWC_{crit2}$ from Eq. 7). The new feature
of BBGCMuSo is that after the optimal soil moisture content, soil stress index can decrease due to saturation soil
stress (anoxic soil).
**4.2 Management modules**
The original Biome-BGC was developed to simulate natural ecosystems with very limited options to disturbance
or human intervention (fire effect is an exception). Lack of management options limited the applicability of
Biome-BGC in croplands and grasslands, but also in managed forests.
One major feature of BBGCMuSo is the implementation of several management options. For grasslands, grazing
and mowing modules were already published as part of the developed Biome-BGC (Hidy et al., 2012).
Since the release of the developed Biome-BGC, we improved the model's ability to simulate management. In
BBGCMuSo, the user can define 7 different events for each management activity, and additionally annually
varying management activities can be defined. Further technical information about using annually varying
management activities is detailed in Section 3.2. of the User's Guide (Hidy et al., 2015). Additionally, new
management modules were implemented and the existing modules were further developed and extended. The
detailed description of mowing and grazing can be found in Hidy et al. (2012).
**4.2.1 Harvest**
In arable crops, the effect of harvest is similar to the effect of mowing in grasslands but the fate of the cut-down
fraction of aboveground biomass is different. We assume that after harvest, snags (stubble) remain on the field as
part of the accumulated biomass, and part of the plant residue may be left on the field (in the form of litter
typically to improve soil quality). Yield is always transported away from the field, while stem and leaves may be
transported away (and utilized e.g. as animal bedding) or may be left at the site. The ratio of harvested
aboveground biomass that is taken away from the field (harvest index) has to be defined as an input.
If residue is left at the site after harvest, the cut-down plant material first goes into a temporary pool that
gradually enters the litter pool. The turnover rate of mown/harvested biomass to litter can be set as an
ecophysiological parameter. Although harvest is not possible outside the growing season, this temporary pool
can contain plant material also in the dormant period (depending on the amount of the cut-down material and the





turnover rate of the pool). The plant material turning into litter compartment is divided between the different
types of litter pools according to the parameterization (based on unstable, cellulose and lignin fractions). The
water stored in the canopy of the cut-down fraction is assumed to be evaporated.

### 4.2.2 Ploughing

As a management practise, ploughing may be carried out in preparation for sowing or following harvest. Three
types of ploughing can be defined in BBGCMuSo: shallow, medium and deep (first; first and second; first,
second and third soil layers are affected, respectively). Ploughing affects the predefined soil texture as it
homogenizes the soil (in terms of texture, temperature and moisture content) for the depth of the ploughing. We
assume that due to the plough the snag or stubble turns into a temporary ploughing pool on the same day. A
fixed proportion of the temporary ploughing pool (an ecophysiological parameter) enters the litter pool on a
given day after ploughing. The plant material turning into the litter compartment is divided between the different
types of litter pools (labile, unshielded cellulose, shielded cellulose and lignin).
A new feature of Biome-BGCMuSo is that aboveground and belowground (buried) litter is handled separately in
order to support future applications of the model in cropland related simulations (presence of crop residues at the
surface affects runoff and soil evaporation). As a consequence of litterfall during the growing season and the
result of harvest, litter accumulates at the surface. In case of ploughing the content of aboveground litter turns
into the belowground litter pool. In this way aboveground/belowground litter amount can be quantified.

### 4.2.3 Fertilization

The most important effect of fertilization in BBGCMuSo is the increase of mineralized soil nitrogen. We define
an actual pool which contains the amount of fertilizer's nitrogen content put out onto the ground on a given
fertilizing day (actual pool of fertilizer; APF). A fixed proportion of the fertilizer enters the top soil layer on a
given day after fertilizing. It is not the entire fraction that enters the soil because a given proportion is leached
(this is determined by the efficiency of utilization that can be set as input parameter). Nitrate content of the
fertilizer (that has to be set by the user) can be taken up by the plant directly; therefore we assume that it goes
into the soil mineral N pool. Ammonium content of the fertilizer (that also has to be set by the user) has to be
nitrified before being taken up by plant, therefore it turns into the litter nitrogen pool. C content of fertilizer turns
into the litter C pool. As a result, APF decreases day-by-day after fertilizing until it becomes empty, which
means that the effect of the fertilization ends (in terms of N input to the ecosystem).

### 4.2.4 Planting

In BBGCMuSo transfer pools are defined to contain plant material as germ (or bud, or non-structural
carbohydrate) in the dormant season from which C and N gets to the normal pools (leaf, stem, and root) in the
beginning of the subsequent growing season. In order to simulate the effect of sowing we assume that the plant
material which is in the planted seed goes into the transfer pools thus increasing its content. Allocation of leaf,
stem and root from seed is calculated based on allocation parameters in the ecophysiological input data. We
assume that a given part of the seed is destroyed before sprouting (this can be adjusted by the user).



### 4.2.5 Thinning

Forest thinning is a new management option in BBGCMuSo. We assume that based on a thinning rate (the proportion of the removed trees), the decrease of leaf, stem and root biomass pools can be determined. After thinning, the cut-down fraction of the aboveground biomass can be taken away or can be left at the site. The rate of transported stem and/or leaf biomass can be set by the user. The transported plant material is excluded from further calculations. The plant material translocated into coarse woody debris (CWD) or litter compartments are divided between the different types of litter pools according to parameterization (coarse root and stem biomass go into the CWD pool; if harvested stem biomass is taken away from the site, only coarse root biomass goes to CWD). Note that storage and transfer pools of woody harvested material are translocated into the litter pool.

The handling of the cut down, non-removed pools differs for stem, root and leaf biomass. Stem biomass (live and deadwood; see Thornton, 2000 for definition of deadwood in case of Biome-BGC) is immediately translocated into CWD without any delay. However, for stump and leaf biomass implementation, an intermediate turnover process was necessary to avoid C and N balance errors caused by sudden changes between specific pools. The parameter "Turnover rate of cut-down non-removed non-woody biomass to litter" (ecophysiological input parameter) controls the fate of (previously living) leaves on cut down trees, and it also controls the turnover rate of dead coarse root (stump) into CWD.

### 4.2.6 Irrigation

In case of the novel irrigation implementation we assume that the sprinkled water reaches the plant and the soil similarly to precipitation. Depending on the amount of the water reaching the soil (canopy water interception is also considered) and the soil type, the water can flow away by surface runoff process while the rest infiltrates into the top soil layer. Irrigation amount and timing can be set by the user.

### 4.2.7 Management related plant mortality

In case of mowing, grazing, harvest and thinning, the main effect of the management activity is the decrease of the aboveground plant material. It can be hypothesized that due to the disturbance related mortality of the aboveground plant material, the belowground living plant material also decreases but at a lower (and hardly measurable) rate. Therefore, in BBGCMuSo we included an option to simulate the decrease of the belowground plant material due to management that affects aboveground biomass. The rate of the belowground decrease to the mortality rate of the aboveground plant material can be set by the user. As an example, if this parameter is set to 0.1, the mortality rate of the belowground plant material is 10% of the mortality of the aboveground material on a given management day. Aboveground material refers to actual pool of leaf, stem and fruit biomass while belowground material refers to actual pool of root biomass and all the storage-transfer pools.

### 4.3 Other plant related model processes

The original Biome-BGC had a number of static features that limited its applicability in some simulations. In Biome-BGC we added new features that can support model applications in a wider context, e.g. in climate




change related studies, modelling exercises comparing free air $CO_2$ enrichment (FACE) experiment data with
simulations (Franks et al., 2013)

### 4.3.1 Phenology

To determine the start and end of the growing season, the phenological state simulated by the model can be used
(White et al., 1999). We have enhanced the phenology module of the original Biome-BGC keeping the original
logic and providing the new method as an alternative. We developed a special growing season index (heat sum
growing season index; HSGSI), which is the extension of the GSI index introduced by Jolly et al. (2005). More
details about HSGSI can be found in Hidy et al. (2012).
In the original, as well as in the modified Biome-BGC model versions, snow cover did not affect the start of the
vegetation period and photosynthesis. We implemented a new, dual snow cover limitation method in
BBGCMuSo. First, the growing season can only start if the snowpack is less than a critical amount (given as mm
of water content stored in the snowpack). Second, the same critical value can also limit photosynthesis during the
growing season (no C uptake is possible above the critical snow cover; we simply assume that in case of low
vegetation, no radiation reaches the surface if snow depth is above a pre-defined threshold). The snow cover
estimation is based on precipitation, mean temperature and incoming shortwave radiation (original model logic is
used here). The critical amount of snow can be defined by the user.

### 4.3.2 Genetically programmed leaf senescence

Besides drought stress related leaf senescence, an optional secondary leaf senescence algorithm was
implemented in BBGCMuSo. For certain crops (e.g. maize and wheat), leaf senescence is the genetically
programmed final stage of leaf development as it is related to the age of the plant tissue in leaves. In crop
models, this process is traditionally determined based on the growing degree-day (GDD) sum. GDD sum is a
measure of heat accumulation to predict plant development stages such as flowering as well as the beginning and
end of grain filling. GDD is calculated using the cumulative difference between mean daily temperature and base
temperature (the latter is an ecophysiological input parameter in BBGCMuSo). In case of planting (i.e., in crop
related simulations), GDD is calculated from the day of planting. In other cases, GDD is calculated from the
beginning of the year. After reaching the predefined GDD threshold leaf mortality rates are calculated based on a
mortality coefficient of genetically programmed leaf senescence (ecophysiological input parameter).

### 4.3.3 New plant pools: fruit and soft stem simulation

In order to enable C and N budget simulation of croplands and explicit yield estimation with BBGCMuSo, fruit
simulation was implemented (which is grain in the case of croplands). Start of the fruit allocation is estimated
based on GDD. After reaching a predefined GDD value (supplied by the user as an input parameter), fruit starts
to grow (allocation is modified). Besides critical GDD there are four additional ecophysiological parameters that
have to be defined by the user related to allocation, C:N ratio of fruit, and labile and cellulose proportion of litter.
In order to support C and N budget simulation of non-woody biomass in BBGCMuSo, a soft stem simulation
was implemented (soft stem is non-photosynthesizing, aboveground, non-woody biomass). Soft stem allocation
is parallel to fine root and leaf allocation (and fruit allocation, if applicable). Four additional ecophysiological



parameters are defined due to soft stem simulation. The primary purpose of our soft stem implementation is to
decrease the overestimation of LAI in herbaceous vegetation calculations. In the original model structure, in the
case of herbaceous vegetation, all aboveground plant material is allocated to leaves, which causes unrealistically
high LAI in many cases. Soft stem allocation (a significant pool in e.g. grasses and crops) enables more realistic
leaf C and LAI values.
**4.3.4 New C4 photosynthesis routine**
In the original and modified Biome-BGC models, $C_4$ photosynthesis was expressed as a sub-version of $C_3$
photosynthesis. It was implemented in a way that only one parameter differed (photons absorbed by
transmembrane protein complex per electron transported; mol mol$^{-1}$) for $C_3$ and $C_4$ plants (Collatz et al., 1991).
Based on the work of Di Vittorio et al. (2010), we implemented a new, enzyme-driven $C_4$ photosynthesis routine
into the photosynthesis module. A new ecophysiological parameter (fraction of leaf N in PEP Carboxylase) was
defined in BBGCMuSo to support the $C_4$ routine. The ecophysiological parameter "fraction of leaf N in
Rubisco" still affects the process of $C_4$ photosynthesis (see Di Vittorio et al., 2010 for details).
**4.3.5 Dynamic response and temperature acclimation of respiration**
Due to the changing environmental conditions, photosynthesis and respiration acclimation and dynamic response
of plants could affect the carbon exchange rates, but this mechanism is missing from many biogeochemical
models (Smith and Dukes, 2012).
In order to implement acclimation and short term temperature dependence of maintenance respiration in
BBGCMuSo, the respiration module was modified in two steps. The maintenance respiration in Biome-BGC is
based on a constant $Q_{10}$ factor. In our approach, first the constant $Q_{10}$ value was modified based on Tjoelker et al.
(2001) who proposed an equation for the short-term temperature dependence of the $Q_{10}$ factor (dynamic
response), and showed how this relationship could improve the accuracy of the modeled respiration. We use the
following equation:
$$Q_{10} = 3.22 - 0.046 * T_{air},$$ (13)
where $T_{air}$ is the daily average air temperature (°C).
This modification results in a temperature optimum at which respiration peaks. The main influence of using a
temperature-dependent $Q_{10}$ value can be detected at high temperature values. In case of a constant $Q_{10}$,
overestimation of respiration can occur at higher temperatures (Lombardozzi et al., 2015).
The model of Tjoelker et al. (2001) is a more realistic method to calculate respiration but it does not take into
account acclimation to the longer term changes in thermal conditions.
Long-term responses of respiration rates to the temperature (i.e., acclimation) were implemented in the second
step based on Atkin et al. (2008) who developed a method using the relationship between leaf respiration, leaf
mass-to-area ratio, and leaf nitrogen content in 19 species of plants grown at four different temperatures. The
proposed equation is the following:
$$R_A = R_T * 10^{A*(T_{10days} - T_{ref})},$$ (14)





where $R_T$ is the non-acclimated rate of respiration, $T_{ref}$ is the reference temperature, $T_{10days}$ is the average daily
temperature in the preceding 10 days, and A is a constant, which was set to 0.0079 based on Atkin et al. (2008).
The acclimated and non-acclimated day and night leaf maintenance respiration functions are presented in Fig. 5.
Photosynthesis acclimation is in a test phase. It is simulated in a very simple way by modifying the relationship
between $V_{cmax}$ (maximum rate of carboxylation) and $J_{max}$ (maximum electron transport rate) that was
temperature-independent in the original code. Temperature dependency is calculated based on average
temperature for the previous 30 days (Knorr and Kattge, 2005). This simple photosynthesis method is not a
complete representation of the acclimation of the photosynthetic machinery (Smith and Dukes, 2012) but a first
step towards a more complete implementation. This feature can be enabled or disabled by the user.
**4.3.6 LAI-dependent albedo**
In order to improve the shortwave radiative flux estimation of BBGCMuSo, an LAI-dependent albedo estimation
was implemented based on the method of Ritchie (1998). Actual albedo is calculated by the following equation:

$$\alpha_{act} = \alpha_{crit} - (\alpha_{crit} - \alpha_{BS}) \cdot \exp(-0.75 \cdot LAI_{act}), \tag{15}$$

where $\alpha_{crit}$ is an empirical estimate of the albedo maximum (0.23), $\alpha_{BS}$ is the albedo of the bare soil that has to be
supplied by the user, and $LAI_{act}$ is the actual value of leaf area index as estimated by the model.
**4.3.7 Stomatal conductance regulation**
Although there are many alternative formulations for stomatal conductance calculation within biogeochemical
models (e.g., Damour et al., 2010), there is no standard method in the scientific community. Many state-of-the-
art ESMs use a variant of the Ball-Woodrow-Berry model (Ball et al., 1987), the Leuning (1995) method, and
the Jarvis (1976) method. There is no proof that one method is significantly better than the other (Damour et al.,

21  2010).

Traditionally, Biome-BGC uses the empirical, multiplicative Jarvis method for stomatal conductance
calculations (Jarvis, 1976). In this method, stomatal conductance is calculated as the product of the maximum
stomatal conductance (model parameter) and limiting stress functions based on minimum temperature, VPD, and
soil water status (Trusilova et al., 2009).
In BBGCMuSo, we kept this logic as its applicability was demonstrated in many studies (see Introduction). In
contrast to the original model, the stress function of soil water status is currently based on relative soil water
content and the introduced SMSI (instead of soil water potential) (Hidy et al., 2012). Beyond this, we modified
the original method of calculation to take into account the changes in ambient $CO_2$ concentration. It was
demonstrated that the stomatal response of Biome-BGC to increasing atmospheric $CO_2$ concentration is not
consistent with observations (Sándor et al., 2015; note the latter study used BBGCMuSo 2.2 for which the
modification was not yet implemented). Therefore, we implemented an additional multiplicative factor in the
calculations in BBGCMuSo 4.0.
In Biome-BGC, stomatal conductance is the function of maximum stomatal conductance (that is an input
ecophysiological parameter) and a set of limitation factors. These multiplicative limitation factors summarize the
effects of SWC, minimum soil temperature, VPD and photosynthetic photon flux density. As we mentioned




above (Section 3.2.1), we modified the limitation function of SWC on stomatal conductance. Besides this
modification, a new $CO_2$ concentration dependent adjustment factor was implemented in order to improve
stomatal calculation based on Franks et al. (2013). They demonstrated a significantly similar dependence of
stomatal conductance on ambient $CO_2$ concentration from hourly to geological time scales. Though the Franks et
al. (2013) paper presents results for stomatal conductance in general (as opposed to maximum stomatal
conductance), we use the assumption that the same function describes changes in maximum stomatal
conductance as well. This assumption is acceptable since stomatal conductance has an upper limit which is
present in conditions without environmental stress and with maximum illumination.
In our approach, data from Franks et al. (2013; Table S1) was used to fit a power function to describe the
quantitative relationship between relative change of stomatal conductance ($g_{w(rel)}$; in fact this is stomatal
conductance to water vapour but this is proportional to stomatal conductance to $CO_2$) to changes of ambient $CO_2$
mixing ratio (Fig. 7b in Franks et al., 2013). The changes are expressed relative to standard conditions:
$$g_{w(rel)} = 39.43 \cdot \left[CO_2\right]^{-0.64} \qquad (16)$$
where $[CO_2]$ is atmospheric $CO_2$ mixing ratio in ppm. In BBGCMuSo v4.0, maximum stomatal conductance is
first re-calculated according to the actual $CO_2$ concentration (this latter is supplied by the user for each
simulation year). As maximum stomatal conductance is no longer constant (which was the case in the original
Biome-BGC and also in BBGCMuSo up to version 3.0), it is assumed that the maximum stomatal conductance
defined by the user represents end of the 20th century conditions (at ~360 ppm mixing ratio representing ~ year
1995 conditions; $g_{smax\_EPC}$).
To calculate actual maximum stomatal conductance, first $g_{w(rel)}$ is calculated at 360 ppm according to Eq. 13
($g_{w(rel)[360ppm]}$=39.43*360$^{-0.64}$=0.9116; note that this value is not exactly one because of the scatter of data used to
fit the power function). Next, $g_{w(rel)}$ is calculated for the given $[CO_2]$ value according to Eq. 16. The final
maximum stomatal conductance ($g_{smax}$) is calculated as:
$$g_{smax} = g_{w(rel)} / 0.9116 \cdot g_{smax\_EPC} \qquad (17)$$
The final maximum stomatal conductance is used for subsequent calculations for evapotranspiration and
photosynthesis. This modification is essential for climate-change related simulations.
**4.4 Modification in the model run**
**4.4.1 Dynamic mortality**
Annual whole-plant mortality fraction (WPM) is part of the ecophysiological parameterization of Biome-BGC
(user supplied value) which is assumed to be constant throughout the simulation. From the point of view of
forest growth, constant mortality can be considered as a rough assumption. Ecological knowledge suggests that
WPM varies dynamically within the lifecycle of forest stands due to competition for resources or due to
competition within tree species (plus many other causes; Hlásny et al., 2014).
In order to enable more realistic forest stand development simulation, we implemented an option for supplying
annually varying WPM to BBGCMuSo (this option can also be used with other biome types with known,
annually varying disturbance level). During the normal phase of the simulation, the model can either use





constant mortality or it can use annually varying WPM defined by the user. Technical details about the
application of this option can be found in Section 3.6 of the BBGCMuSo v4.0 User's Guide (Hidy et al., 2015).
**4.4.2  Optional transient run**
Model spinup in Biome-BGC typically represents steady-state conditions before the industrial revolution
(without changing atmospheric $CO_2$ concentration and N deposition). Therefore, the usual strategy for spinup is
to use constant (preindustrial) $CO_2$ and $N_{dep}$ values during the spinup followed by annually varying values for the
entire normal simulation (representative to present day conditions or to 20[th] century conditions). This strategy
might be used due to unknown site history or a lack of driving data. However, this logic can lead to undesired
transient behavior of the model results as the user may introduce a sharp change by the inconsistent $CO_2$ and/or
$N_{dep}$ data between the spinup and normal phase (note that both $CO_2$ and $N_{dep}$ are important drivers of plant
growth).
In order to avoid this phenomena (and more importantly, to take into account site history), some model users
performed one or more transient simulations to enable smooth transition from one simulation phase to the other
(mainly, from the spinup phase to the normal phase; Thornton et al., 2002; Vetter et al., 2005; Hlásny et al.,
2014). However, this procedure means that the users have to perform a 3rd model run (and sometimes even
more), using the output of the spinup phase, and create the input that the normal phase can use.
In BBGCMuSo, we implemented a novel approach to eliminate the effect of sharp changes in the environmental
conditions between the spinup and normal phase. According to the modifications, now it is possible to make an
automatic transient simulation after the spinup phase.
To initiate the transient run, the user can adjust the settings of the spinup run. If needed, a regular spinup will
first be performed with constant $CO_2$ and N deposition values followed by a second run to be performed using
the same meteorological time series defined for the spinup. In this way, the length of the transient run is always
equal to the length of the meteorology data used for the spinup phase. During the transient run, annually varying
$CO_2$ concentration data can be used (but it should be constructed to provide transition from preindustrial to
industrial $CO_2$ concentrations). Utilization of annually varying N-deposition data is optional but it is preferred.
The input data of transient run is the output of the spinup phase, and the output of the transition run is the input
of the normal phase. Technical details about the transient run can be found in Section 2.2 of the User's Guide
(Hidy et al., 2015).
As management might play an important role in site history (and consequently in biogeochemical cycles), the
new transient simulation in BBGCMuSo can include management in an annually varying fashion (management
is described below).
**4.4.3  Land-use-change simulations**
Another new feature was added to BBGCMuSo that is also related with the proper simulation of site history. As
the spinup phase is usually associated with preindustrial conditions, the normal phase might represent a plant
functional type that is different from the one present in the spinup phase. For example, present-day croplands can
occupy land that was originally forest or grassland, so in this case spinup will simulate forest in equilibrium, and
then the normal phase will simulate croplands. Another example is the simulation of afforestation that might



require spinup for grasslands, and normal phase for woody vegetation. We may refer to these scenarios as land
use change (LUC) related simulations.
One problem that is associated with LUC is the frequent crash of the model with the error 'negative nitrogen
pool' during the beginning of the normal phase. This error is typical if the spinup and normal ecophysiological
parameterization differ in terms of plant C:N ratios (due to the internal model logic; inconsistency might arise
between available mineralized N and N demand by the plant, and this might cause a negative nitrogen pool).
In order to avoid this error, we have implemented the following automatic procedure. According to the changes,
only the equilibrium C pools are passed to the normal phase after the spinup phase has ended. The equilibrium
nitrogen pools are calculated by the model code so that the resulting C:N ratios are harmonized with the C:N
ratio of the different plant compartments presented in the ecophysiological parameterization of the normal phase.
This modification means that equilibrium nitrogen pools are not passed to the normal phase. We believe this
issue is compensated with the fact that LUC and site history can be simulated properly.

### 4.5 Empirical estimation of other greenhouse gases

Quantification of the full greenhouse gas (GHG) balance of ecosystems was not possible with the original
Biome-BGC. Non-$CO_2$ greenhouse gases (GHGs) are important elements of the biogeochemistry of the soil-
plant system. Nitrous oxide ($N_2O$) and methane ($CH_4$) are strong GHGs that can eventually compensate the $CO_2$
sink capacity of ecosystems resulting in GHG-neutral or GHG source activity (Schulze et al., 2009). We
established the first steps towards improved modeling possibilities with Biome-BGC that includes soil efflux
estimation of non-$CO_2$ GHGs.

#### 4.5.1 Estimation of nitrous oxide and methane flux of unmanaged soil

Due to their importance, an empirical estimation of $N_2O$ and $CH_4$ emission from soils was implemented in
BBGCMuSo based on the method of Hashimoto el al. (2011). Gas fluxes are described in terms of three
functions: soil physiochemical properties (C:N atio for $N_2O$; bulk density of top soil layer for $CH_4$), water-filled
pore space (WFPS) of top soil layer, and soil temperature of top soil layer.
The Hashimoto et al. (2011) method was developed for unmanaged ecosystems. We adapted it to managed
ecosystems by including emission estimates from livestock and manure management using Tier 1 methods of
IPCC (2006).

#### 4.5.2 Estimation of nitrous oxide and methane flux from grazing and fertilizing

The IPCC (2006) Tier 1 method was implemented in BBGCMuSo to give a first estimation to $CH_4$ and $N_2O$
emissions due to grazing and fertilizing.
Due to their large population and high $CH_4$ emission rate, cattle are an important source of $CH_4$. Methane
emissions of cattle originate from manure management (including both dung and urine) and enteric emissions.
Methane emission from enteric fermentation is estimated using the number of the animals and the regional-
specific methane emission factor from enteric fermentation (IPCC, 2006; Chapter 10, Table 10.11). Another
mode of methane production is the emission during the storage of manure. Methane emission from manure



management is estimated using the number of animals and the regional-specific methane emission factor from
manure management (IPCC, 2006; Chapter 10, Table 10.14).
In summary, the flux of the methane ($F_{CH4}$; mg $CH_4$ m$^{-2}$ day$^{-1}$) is the sum of soil flux, flux from fermentation and
from manure management:

$$\left(F_{CH_4}\right)_{total} = \left(F_{CH_4}\right)_{soil} + \left(F_{CH_4}\right)_{fermentation} + \left(F_{CH_4}\right)_{manure} \qquad (18)$$

The emissions of $N_2O$ that result from anthropogenic N inputs or N mineralization occur through both direct
pathways (directly from the soils to/from which the N is added/released – grazing and fertilization), and through
indirect pathways (volatilization, biomass burning, and leaching). Volatilization fluxes were already
implemented into BBGCMuSo. Direct emissions of $N_2O$ from managed soils consist of the emission from
animal excretion and from fertilization. The former is estimated based on multiplying the total amount of N
excretion by an emission factor for that type of manure management system (IPCC, 2006; Chapter 10, Table
10.21). The latter is estimated using emission factors developed for $N_2O$ emissions from synthetic fertilizer and
organic N application (IPCC, 2006; Chapter 11, Table 10.21).
In summary, the flux of the nitrous oxide ($F_{N2O}$; mg $N_2O$ m$^{-2}$ day$^{-1}$) is the sum of soil flux, flux from
fermentation and from manure management:

$$\left(F_{N_2O}\right)_{total} = \left(F_{N_2O}\right)_{soil} + \left(F_{N_2O}\right)_{grazing} + \left(F_{N_2O}\right)_{fertilizing} \qquad (19)$$

**5 Simulation results: model evaluations**
In order to examine and evaluate the functioning of BBGCMuSo, case studies are presented in this section
regarding different vegetation types: $C_3$ grassland (Bugac, Hungary), $C_4$ maize (Mead, Nebraska, United States
of America) and $C_3$ oak forest (Jastrebarsko, Croatia).
The model behavior is evaluated by visual comparison of measured and simulated data and by quantitative
measures such as root mean squared error (RMSE), normalized root mean squared error (RMSE weighted by the
difference of maximum and minimum of the measured data), bias (average difference between simulated and
measured variables, where positive bias means systematic overestimation and negative bias means overall
underestimation), coefficient of determination ($R^2$) of the regression between measured and modeled data, and
Nash–Sutcliffe modeling efficiency (NSE). The range of NSE is between 1.0 (perfect fit) and $-\infty$. An efficiency
of lower than zero means that the mean of the observed time series would have been a better predictor than the
model. NSE and $R^2$ are dimensionless. The dimension of RMSE and bias is the dimension of the variable to
which it refers. The dimension of NRMSE is %.
In this study, model parameters were taken from the literature, and also from previous parameterization of
Biome-BGC. For the newly introduced parameters, we use values that provided reasonable results during model
development. The number of parameters in the original model (Biome-BGC v4.1.1 MPI) and BBGCMuSo
(Biome-BGCMuSo v4.0) is not the same (MuSo has more ecophysiological parameters than original model,
while a few parameters of the original model are no longer used in BBGCMuSo) but the same values were set in
case of common parameters.





The simulation results of the original Biome-BGC and BBGCMuSo are compared to the measurements.
Different model features such as the senescence mechanism, management, and groundwater effects are
illustrated in the Supplementary material (S1-S9).
BBGCMuSo has more than 600 different output variables. In this study, we focus on the variables relevant for
the evaluation of model performance: that are GPP, TER, net ecosystem exchange (NEE), LHF, LAI, C content
of aboveground biomass (abgC), and SWC in the 10-30 cm soil layer.

## 5.1 C3 grassland/pasture

Measured data at Bugac are available from 2003 to 2015. In order to present model behavior, only three
consecutive years were selected (2009-2011) that represented different meteorological conditions. In 2009, the
annual sum of precipitations was 14% lower, and mean annual temperature was 10% higher than the long term
mean. In 2010, precipitation was 65% higher and temperature was near average (2% lower). In 2011,
precipitation was 22% lower than average and temperature was close to the long term mean.
For the Bugac simulation, several novel features were used from BBGCMuSo v4.0. The HSGSI-based
phenology, transient simulation, drought related plant senescence, standing dead biomass pool, grazing settings
(see Hidy et al. (2015) for grazing related input data), non-zero soft stem allocation, management-related
decrease of storage and the actual pool, and acclimation were all used in the simulation.
Both during spinup and normal phase of the simulation we assumed that the vegetation type is grassland,
therefore the $C_3$ grass parameterization was used (see Supplementary material Table S1 for parameterization).
Fig. 6 shows that the original Biome-BGC overestimates GPP, TER, LAI and LHF. Due to the implementation
of grazing and plant senescence processes, the overestimation of GPP, TER and LHF were decreased at the
Bugac site where drought frequently occurs. The average observed maximum LAI is about 4-5 $m^2$ $m^{-2}$ at Bugac.
Therefore, due to model improvements simulated LAI is more realistic using BBGCMuSo. This was possible
due to the implementation of the new soft stem pool that now can contain part of the aboveground C (in the
original model, all aboveground C formed leaf biomass for grasses).
The results of the quantitative model evaluation are presented in Table 3 for Bugac using the original Biome-
BGC and the BBGCMuSo. The statistical indicators show that with the exception of SWC, the simulation
quality typically improved due to the model developments. The error metrics are usually smaller, $R^2$ is higher
than in case of the original model using the original parameter set. BBGCMuSo-simulated LHF has higher bias
(in absolute sense) than the original model, but the other error metrics perform better. It is notable that NSE
became positive for GPP, TER and LHF while it was negative for all three variables with the original Biome-
BGC.
The SWC estimation by the original model was closer to the observations than the result with BBGCMuSo.
However, as the original Biome-BGC has a simple, one-layer bucket module for soil hydrology, SWC estimation
by the original model and topsoil SWC simulation by Biome-BGCMuSo are not comparable. SWC provided by
the original model has no vertical profile, which means that it is not expected that the simulation will match
observation. Nevertheless, although the errors are higher in case of BBGCMuSo (RMSE, NRMSE, NSE, BIAS),
the correlation is higher. This is relevant because in the BBGCMuSo simulation, the relative change of soil water



content significantly impacts the senescence and decomposition fluxes (see Section 3.2). Therefore, if the
relative change of the soil water content is realistic, the soil moisture limited processes can still be realistic in
spite of the obvious bias in the measurements.
Supplementary material S2 contains additional simulation results for Bugac demonstrating the effect of the long
lasting drought on plant state, the senescence effect on plant processes and carbon balance and the effect of
grazing.
**5.2 C4 cropland**
Measured data and simulated results are presented for the 2003-2006 period for the Mead1 site (Fig. 7).
For Mead, the applied novel features with BBGCMuSo included $C_4$ enzyme-driven photosynthesis, transient
simulation, planting, harvest, irrigation, drought related plant senescence, genetically programmed leaf
senescence, standing dead biomass pool, fruit pool (in this case maize yield), non-zero soft stem allocation, and
acclimation. Note that the allocation related "current growth proportion" parameter that controls the content of
the storage pool (non-structured carbohydrate reserve for next year's new growth) was zeroed here in order to
avoid natural plant growth in the spring. In other words, in case of planting, the storage pool has to be turned off
as maize growth is only possible after sowing.
During the spinup phase, we assumed that the vegetation type was grassland before cropland establishment.
Therefore $C_3$ grass parameterization was used. In the normal phase, maize parameterization was used. Note that
change in ecophysiological parameterization was possible due to the developments (see Section 2).
Supplementary material Table S1 contains the complete parameterization for the maize simulation.
As noted (Section 3.5.2), there is a high chance for model error if the spinup and normal ecophysiological
parameterization differ in terms of plant C:N ratios (due to the internal model logic). To avoid this error, in the
case of original Biome-BGC, the C:N ratios of maize were used in the spinup phase. Mead1 is under
management including irrigation, fertilization, harvest, and ploughing. The original model assumes that the
vegetation is undisturbed, so no management practices could be set. For BBGCMuSo, management was set
according to information taken from the site PI.
Fig. 7 shows that the original model underestimated GPP in the vegetation period (in contrast to BBGCMuSo)
because it cannot take into account the effect of irrigation and fertilization, which support the growth of
agricultural crops. The original model overestimated GPP following harvest because plant material was not
removed from the site. Using BBGCMuSo, the overestimation of TER and LAI decreased compared to the
original model due to the simulated harvest. For BBGCMuSo this overestimation decreased also due to the novel
fruit and soft stem simulations (in the original model, aboveground plant material apportions biomass to leaves
only).
The results of the quantitative model evaluation are presented in Table 4 for Mead using the original Biome-
BGC and the BBGCMuSo. According to the error metrics the simulation quality improved due to the model
developments for GPP, TER, LAI and abgC, and also for SWC. For BBGCMuSo, the errors (both RMSE and
bias) are smaller and square of correlations are higher than in case of the original model using the original
parameters. Values of NSE are close to 1.0 using BBGCMuSo indicating a good match of the modeled values
and the observed data. For the original model, NSE values are much smaller or negative. NSE is negative for



SWC both for the original and the BBGCMuSo version in spite of the lower bias of the latter. Again, due to the very simplistic SWC module of the original model, its SWC results are not directly comparable with observations.

Supplementary material S3 presents additional simulation results for the Mead1 site demonstrating the effect of new model developments like irrigation fertilization, and the genetically programmed leaf senescence.

**5.3 Deciduous broad-leaved forest**

For Jastrebarsko forest site measurement data were available for the period from 2008 to 2014. But, as the measurement height was changed in spring 2011 (flux tower was upgraded due to tree growth), to assure the homogeneity of the measurement data we used data from 2008-2010 only.

For simulation at Jastrebarsko, several novel features were used from BBGCMuSo v4.0. For example, transient simulation, alternative tipping bucket soil water balance calculation, groundwater effect on stomatal conductance and decomposition, drought related plant senescence, thinning, fruit allocation, bulk denitrification, soil moisture limitation calculation, soil stress index, soil texture change through the soil profile, and respiration acclimation were all used in the simulation. We also used the possibility to adjust the parameter for maintenance respiration, which is fixed within the source code in the original Biome-BGC model. We used the value of 0.4 kg C kg $N^{-1}$ $day^{-1}$ based on newly available data (Cannell and Thornley, 2000).

During both spinup and normal phase we assume that the vegetation type was pedunculate oak forest, therefore oak parameterization was used (see Supplementary material Table S1 for parameterization).

Fig. 8 shows that the original Biome-BGC overestimates GPP, TER, NPP and LAI. Probable reason for this is high N fixation parameter value, estimated to 0.0036 g N $m^{-2}$ $yr^{-1}$, resulting from the presence of N-fixer species *Alnus glutionsa* (see Supplement for details) and, at the same time, the lack of denitrification process in the original Biome-BGC version. Due to the implementation of dry and wet bulk denitrification processes, effects of groundwater, and soil moisture stress in the BBGCMuSo version the overestimation of GPP, TER, and NPP was no longer present and simulation results significantly improved. Jastrebarsko forest site is a lowland oak forest with complex soil hydrology which cannot easily be simulated using simple soil bucket approach, as it is a case in original Biome-BGC model. Soil at Jastrebarsko site is stagnic luvisol, with impermeable clay horizon at 2-3 m depth and poor vertical water conductivity, resulting with parts of the forest being partly waterlogged or even flooded with stagnating water during winter and early spring. Flooding leads to increased denitrification (Groffman and Tiedje, 1989; Kulkarni et al., 2014) and, in combination with hypoxic conditions in the root zone, reflects negatively on GPP and TER (Fig. 8a,b).

Soil water content simulation in BBGCMuSo version shows improvement for the leaf-off season (Nov-Apr) when, unlike the original version, soil water saturation in BBGCMuSo simulation becomes evident (Fig. 8c). BBGCMuSo still overestimates soil moisture status in the summer. According to the original Biome-BGC the forest never experiences soil saturation which is contrary to the actual situation in winter and early spring. Original Biome-BGC provides only SWC data for one layer of the entire rooting zone (0-100 cm in this case) and therefore it cannot be directly compared with the actual measurements or output from BBGCMuSo which correspond to 0-30 cm soil layer. Also, it should be emphasized that the parameters soil bulk density, SWC of



saturation, field capacity, and wilting point for soil layers were not available. BBGCMuSo calculated those
parameters based only on the provided parameters for soil texture.
Simulated LAI value also decreased to more realistic values in BBGCMuSo due to model improvements (Fig.
8d). Based on litterfall data (0.408 kg m$^{-2}$; Marjanović et al. (2011b)) and average specific leaf area of 15 m$^2$ kg$^{-1}$
for oak (Morecroft and Roberts, 1999), the average maximum LAI at Jastrebarsko is estimated to be 6.1 m$^2$ m$^{-2}$,
just slightly above what is estimated by BBGCMuSo. BBGCMuso NPP estimates are also in good agreement
with those from field observation (Marjanović et al., 2011a), unlike those of the original version which
overestimate NPP (Fig. 8e)
The results of the quantitative model evaluation are presented in Table 5 for Jastrebarsko site using the original
Biome-BGC and the BBGCMuSo. The quality of simulation usually improved with BBGCMuSo due to the
model developments ($R^2$ for SWC is an exception, but see the notes above about the SWC simulation of the
original Biome-BGC). Key variables' errors and biases are smaller in comparison to the original model using the
original parameter set.
Supplementary material S4 contains additional simulation results for the Jastrebarsko site demonstrating the
effect of new model developments.
**6 Discussion and conclusions**
Biogeochemical models inevitably need continuous improvement to improve their ability to simulate ecosystem
C balance. In spite of the considerable development in our understanding of the C balance of terrestrial
ecosystems (Baldocchi et al., 2001; Friend et al., 2006; Williams et al., 2009), current biogeochemical models
still have inherent uncertainties. The major sources of uncertainties are internal variability, initial and boundary
conditions (e.g. equilibrium soil organic matter pool estimation), parameterization and model formulation
(process representation) (Schwalm et al., 2015; Sándor et al., 2016).
In this study, our objective was to address model structure related uncertainties in the widely used Biome-BGC
model.
The interactions between water availability and ecosystem C balance are widely documented in the literature. In
a recent study, Ahlström et al. (2015) showed that semi-arid ecosystems have strong contribution to the trend
and interannual variability of the terrestrial C sink. This finding emphasizes the need to accurately simulate soil
water related processes and plant senescence due to prolonged drought. In earlier model versions, drought could
not cause plant death (i.e., LAI decrease) which resulted in a quick recovery of the vegetation after a prolonged
drought. In our implementation, long lasting drought can cause irreversible plant senescence which is more
realistic than the original implementation.
Role of management as the primary driver of biomass production efficiency in terrestrial ecosystems was shown
by Campioli et al. (2015). The study clearly demonstrates that management cannot be neglected if proper
representation of the ecosystem C balance is needed. The management modules built into BBGCMuSo cover, on
a global basis, the majority of human interventions in managed ecosystems. As management practises require
prescribed input data for BBGCMuSo, accuracy of the simulations will clearly depend on the quality of
management description which calls for proper data to drive the model.

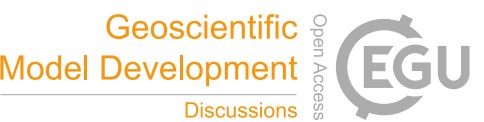

Representation of acclimation of plant respiration and photosynthesis was shown to be one major uncertainty in
global biogeochemical models (Lombardozzi et al., 2015). We introduced acclimation of autotrophic respiration
in BBGCMuSo which is a major step towards more realistic climate change related simulations. Representation
of photosynthesis acclimation (Medlyn et al., 2002) is needed in future modifications as another major
development.
Due the implementation of many novel model features, the model logic became more complex. Thus, process
interactions became even more complicated which means that extensive testing is needed to find problems and
limitations.
The earlier version of BBGCMuSo (v2.2) was already used in a major model intercomparison project (FACCE
MACSUR – Modelling European Agriculture with Climate Change for Food Security, a FACCE JPI knowledge
hub; Ma et al., 2014; Sándor et al., 2016). Within MACSUR, nine grassland models were used and their
performance was tested against EC and biomass measurements. BBGCMuSo v2.2 results were comparable with
other models like LPJmL, CARAIB, STICS, EPIC, PaSim, and others. In spite of the successful application of
the predecessor of BBGCMuSo v4.0, SWC simulations were still problematic (Sándor et al., 2016).
Developments are clearly needed in terms of soil water balance and ecosystem scale hydrology in general.
Other developments are still needed to improve simulations with dynamic C and N allocation within the plant
compartments (Friedlingstein et al., 1999; Olin et al., 2015). Complete representation of ammonium and nitrate
pools with associated nitrification and denitrification is also needed to avoid ill-defined, N balance related
parameters (Thomas et al., 2013).
In the present paper three case studies were presented to demonstrate the effect of model structural
improvements on the simulation quality. Clearly, extensive testing is required at multiple EC sites to evaluate the
performance of the model and to make adjustments if needed.
Parameter uncertainty also needs further investigations. Model parameterization based on observed plant traits is
possible today thanks to new datasets (e.g., White et al., 2000; Kattge et al., 2011; van Bodegom et al., 2012).
However, in some cases, imperfect model structure may cause distortion owing to compensation of errors
(Martre et al., 2015). Additionally, plant traits (e.g. specific leaf area, leaf C:N ratio, photosynthesis-related
parameters) vary considerably in space so even though plant trait datasets are available they might not capture
the site-level parameters. These issues raise the need for proper infrastructure for parameter estimation (or in
other words, calibration or model inversion). The need for model calibration is also emphasized as we introduced
a couple of empirical parameters in BBGCMuSo (e.g. those related to plant senescence, water stress thresholds
and denitrification drivers, disturbance-related mortality parameters) that need optimization using measurement
data from multiple sites. In other words, though BBGCMuSo parameterization is mostly based on observable
plant traits, model calibration can not be avoided in the majority of the cases (Hidy et al., 2012).
To address this problem we created a computer-based open infrastructure that can help a wide array of users in
the application of the new model. We followed the concept of model-data fusion (MDF; Williams et al., 2009)
when developing the so-called "Biome-BGC Projects Database and Management System" (BBGCDB;
http://ecos.okologia.mta.hu/bbgcdb/) and the BioVeL Portal (http://portal.biovel.eu) as a virtual research
environment and collaborative tool. BBGCDB and the BioVeL portal help parameter optimization of
BBGCMuSo by a computer cluster-based Monte Carlo experiment and GLUE methodology (Beven and Binley,



1992). Further integration is planned using an alternative a Bayesian calibration algorithms (Hartig et al., 2012),
that will provide extended calibration options to the parameters of BBGCMuSo.

We would like to support BBGCMuSo application for the wider scientific community as much as possible. For
users of the original Biome-BGC, the question is of course the following: should I move to BBGCMuSo? Does
it need a long learning process to move to the improved model? We would like to stress that users of the original
Biome-BGC will have a smooth transition to BBGCMuSo, as we mainly preserved the structure of the input
files. The majority of changes are related to the 7-layer soil module and to the implementation of management
modules. The ecophysiological parameter file is similar to the original but it was extended considerably. This
may raise issues but as it is emphasized in our User's Guide (Hidy et al., 2015), we extended the
ecophysiological parameter file with many parameters that will not need adjustment in the majority of the cases.
We simply moved some 'burned-in' parameters into the ecophysiological parameter in order to allow easy
adjustment if needed in the future.

**Code availability**

The source code, the Windows model executable, sample simulation input files and documentation are available
at the BBGCMuSo website (http://nimbus.elte.hu/bbgc).

**Authors' Contributions**

Hidy developed Biome-BGCMuSo with modifying Biome-BGC 4.1.1 MPI version. The study was conceived
and designed by Hidy and Barcza, with assistance from Marjanović, Churkina, Fodor, Horváth, and Running. It
was directed by Barcza and Hidy. Marjanović, Ostrogović Sever, Pintér, Nagy, Gelybó, Dobor and Suyker
contributed with simulations, measurement data and its post-processing and ideas on interpretation of the model
validation. Hidy, Barcza and Marjanović prepared the manuscript and the supplement with contributions from all
co-authors. All authors reviewed and approved the present article and the supplement.



**Acknowledgements**
The research was funded by the Hungarian Academy of Sciences (MTA PD 450012), the Hungarian Scientific
Research Fund (OTKA K104816), the BioVeL project (Biodiversity Virtual e-Laboratory Project, FP7-
INFRASTRUCTURES-2011-2, project number 283359), the EU FP7 WHEALBI Project (Wheat and Barley
Legacy for Breeding Improvement; project No. 613556), the Croatian Science Foundation Installation Research
Project EFFEctivity (HRZZ-UIP-11-2013-2492), and metaprogramme Adaptation of Agriculture and Forests to
Climate Change (AAFCC) of the French National Institute for Agricultural Research (INRA). We acknowledge
the international research project titled "FACCE MACSUR – Modelling European Agriculture with Climate
Change for Food Security, a FACCE JPI knowledge hub". Biome-BGC version 4.1.1 (the predecessor of
BBGCMuSo) was provided by the Numerical Terradynamic Simulation Group (NTSG) at the University of
Montana, Missoula MT (USA), which assumes no responsibility for the proper use by others. We are grateful to
the Laboratory of Parallel and Distributed Systems, Institute for Computer Science and Control (MTA SZTAKI),
that provided consultation, technical expertise and access to the EDGeS@home volunteer desk top grid system
in computationally demanding analysis.



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





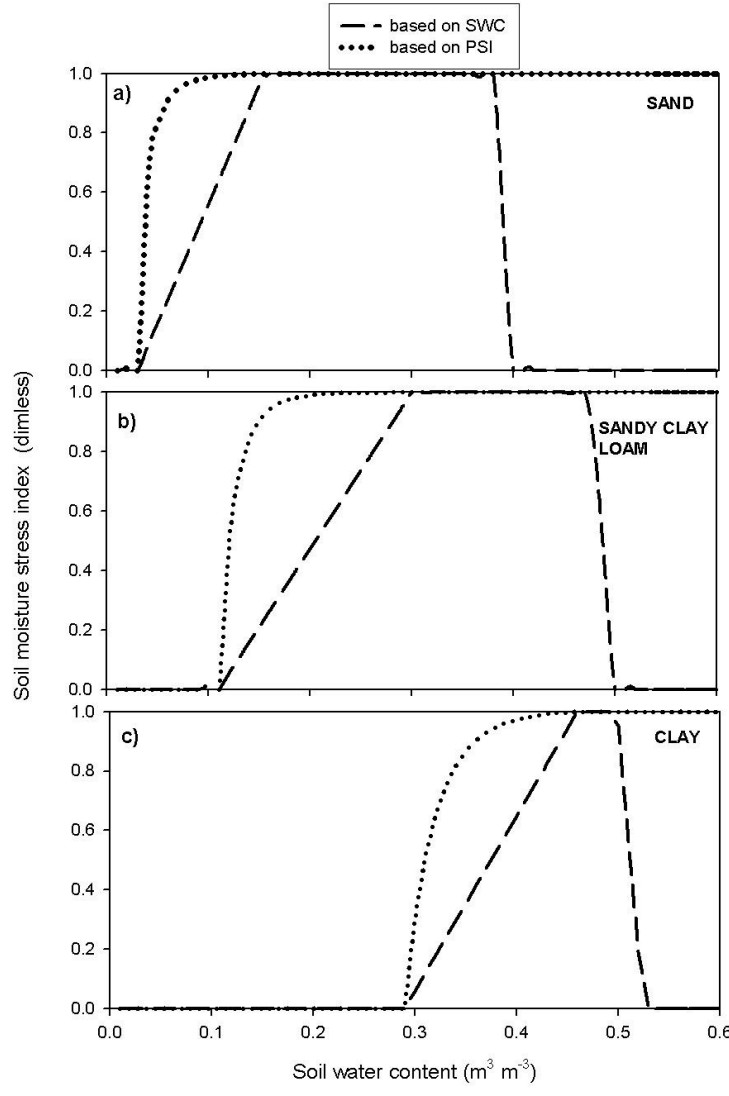

Figure 2. Dependence of the soil moisture stress index on soil water content for three different soil types: sand (a),
sandy clay soil (b) and clay (c). Soil stress index based on soil water potential (dotted lines) is used by the original
model, while soil stress index based on normalized soil water content is used by BBGCMuSo (dashed lines).

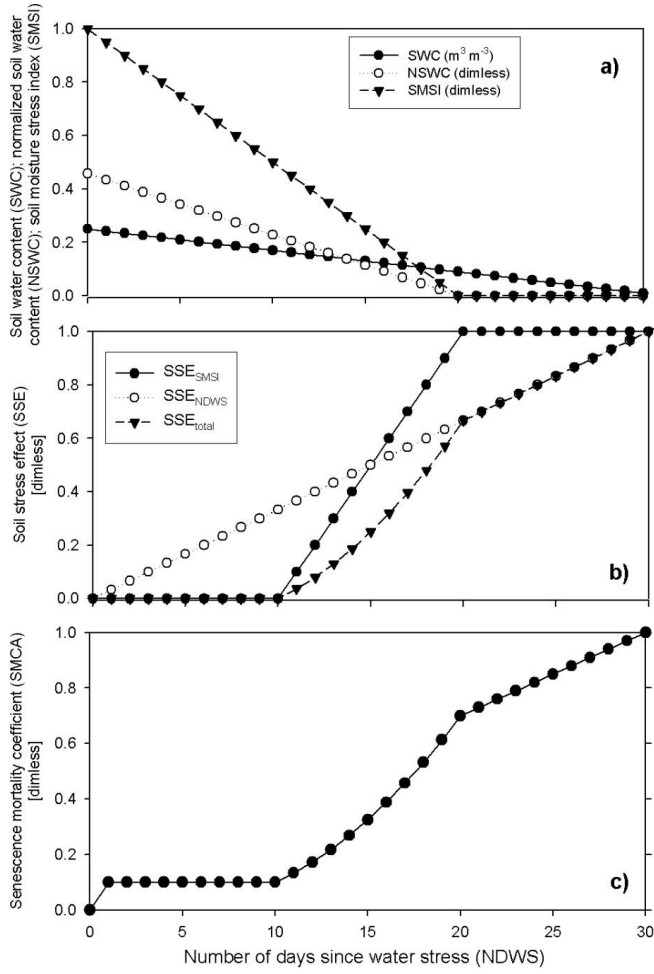



**Figure 3. Demonstration of the senescence calculation with an example. a) Soil water content (SWC; black dots), normalized soil water content (NSWC; white dots) and soil moisture stress index (SMSI; black triangles) during 30 days of a hypothetical drought event. b) Soil stress effect based on soil moisture stress index (SSE$_{SMSI}$; black dots), soil stress effect based on number of days since water stress (SSE$_{NDWS}$) and total soil stress effect (SSE$_{total}$). c) Aboveground senescence mortality coefficient (SMCA).**





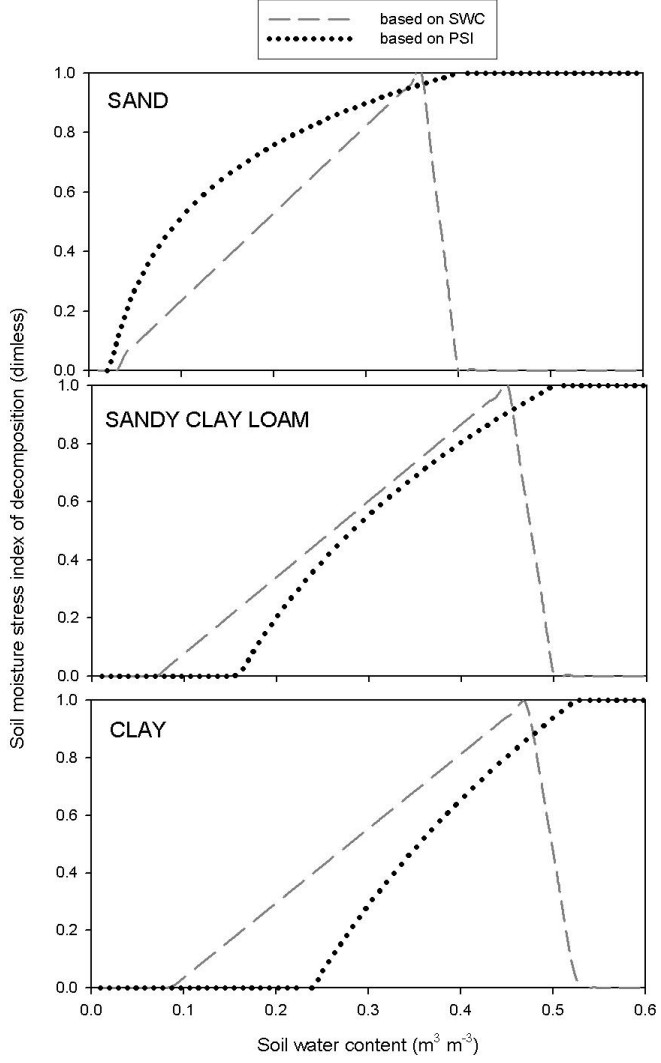

2 **Figure 4. Soil moisture stress index used by decomposition for three different soil types: sand (a), sandy clay soil (b)**
3 **and clay (c). Soil stress index based on soil water potential (dotted lines) is used by the original model, while soil stress**
4 **index based on soil water content is used by BBGCMuSo (dashed grey lines).**





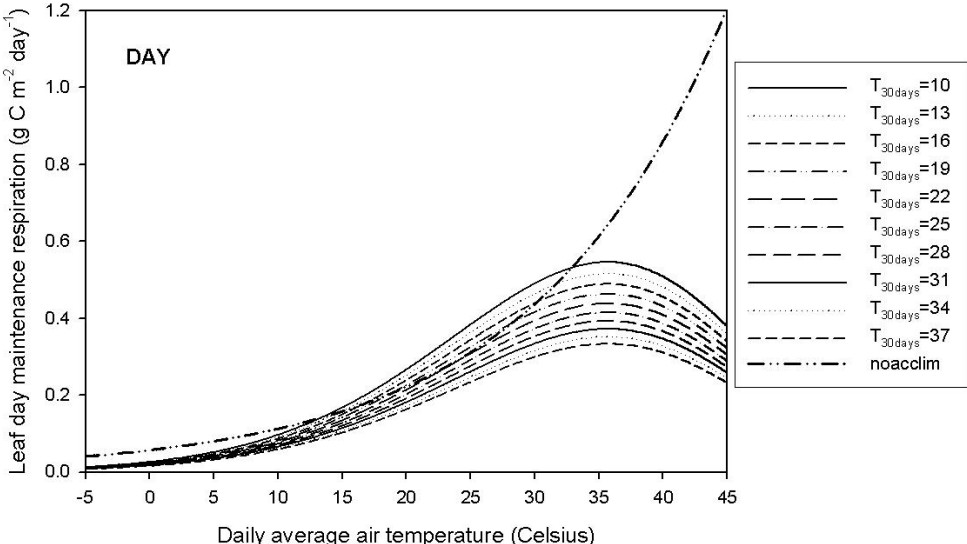

**Figure 5. Acclimated and non-acclimated (noacclim) daytime maintenance respiration of leaf as a function of air**
**temperature. Acclimated respiration values were calculated based on different average daily temperatures during the**
**previous 30 days (T$_{30days}$: 10, 13, 16, 19, 22, 25, 28, 31, 34, and 37 Celsius). In case without acclimation, the respiration**
**is independent of average daily temperature, therefore only one line is presented (this refers to the original model**
**logic).**





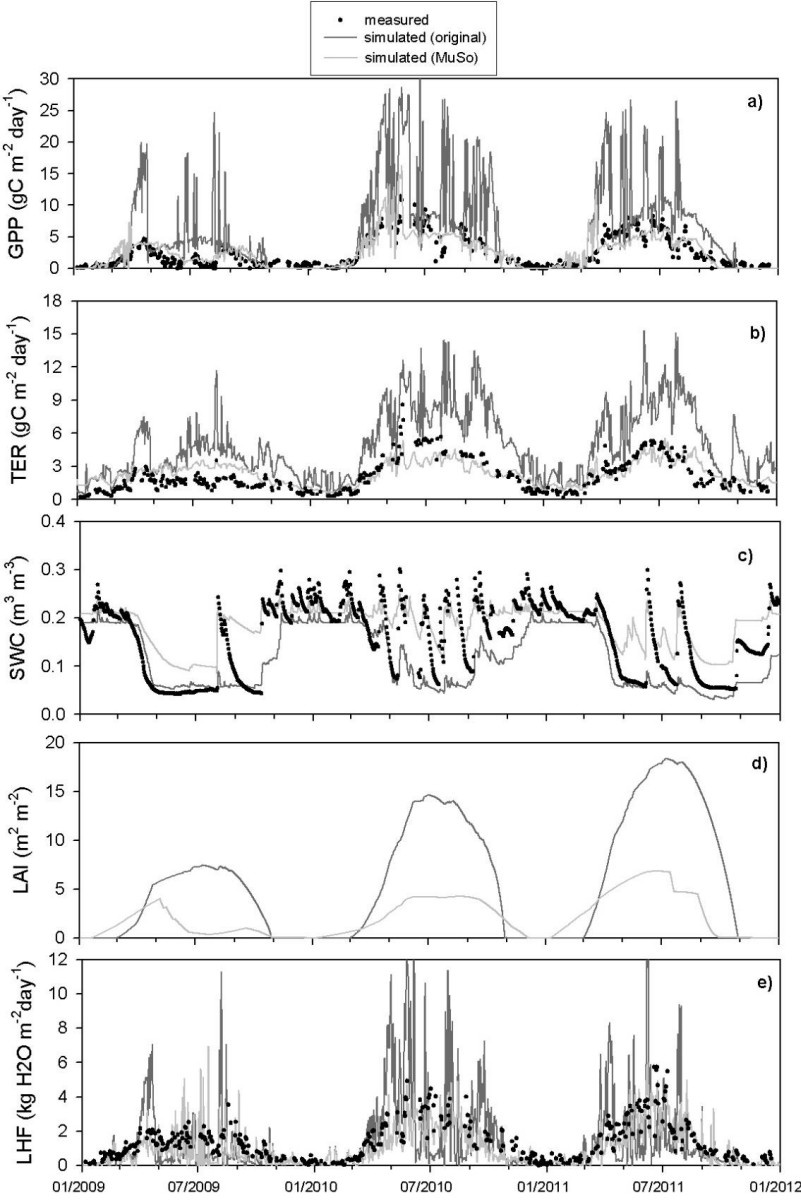

**Figure 6. Measured (black dots) and simulated variables (a) GPP, b) TER, c) SWC, d) LAI, e) LHF) using the original Biome-BGC (dark grey lines) and BBGCMuSo (light grey lines) models for grassland at the Bugac site between 2009 and 2011. Measured LAI was not available at Bugac.**





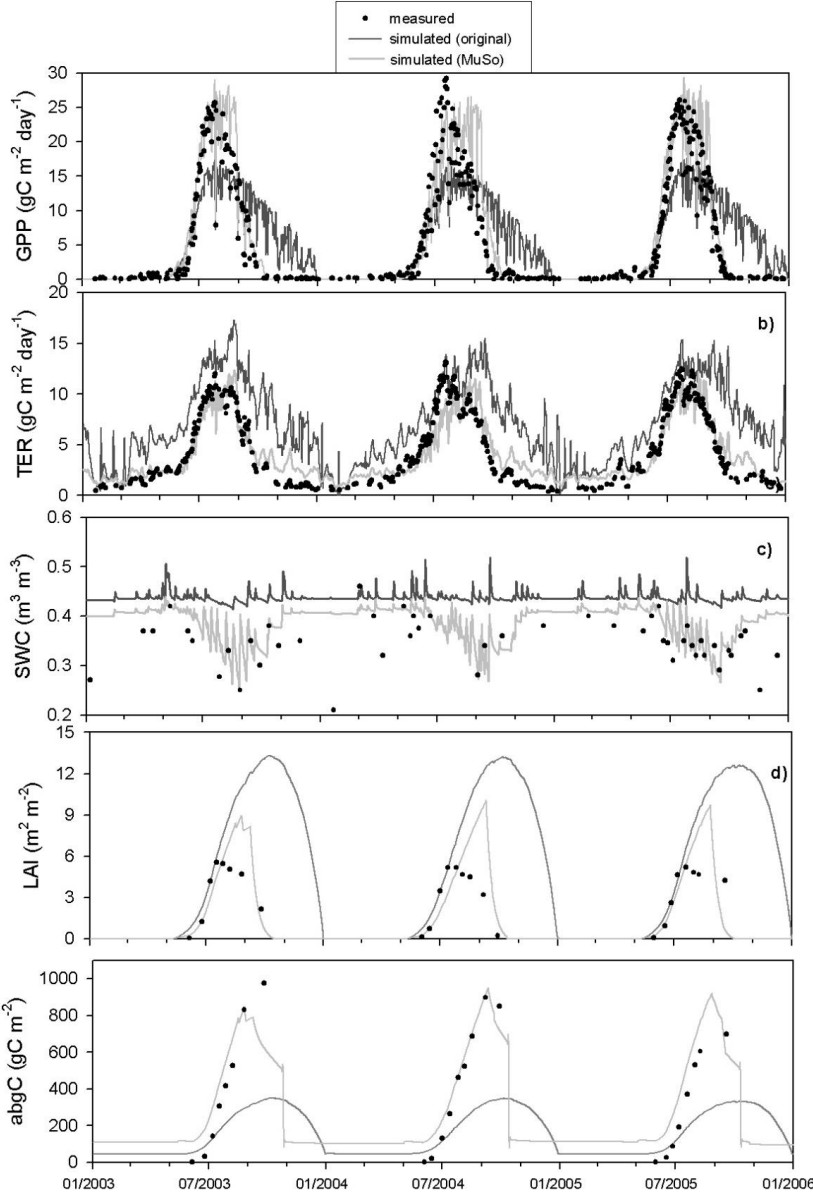

2   **Figure 7. Measured (black dots) and simulated variables (a) GPP, b) TER, c) SWC, d) LAI, e) abgC using original**
3   **Biome-BGC (dark grey lines) and BBGCMuSo (light grey lines) models for maize simulation at the Mead1 site,**
4   **between 2003 and 2006.**





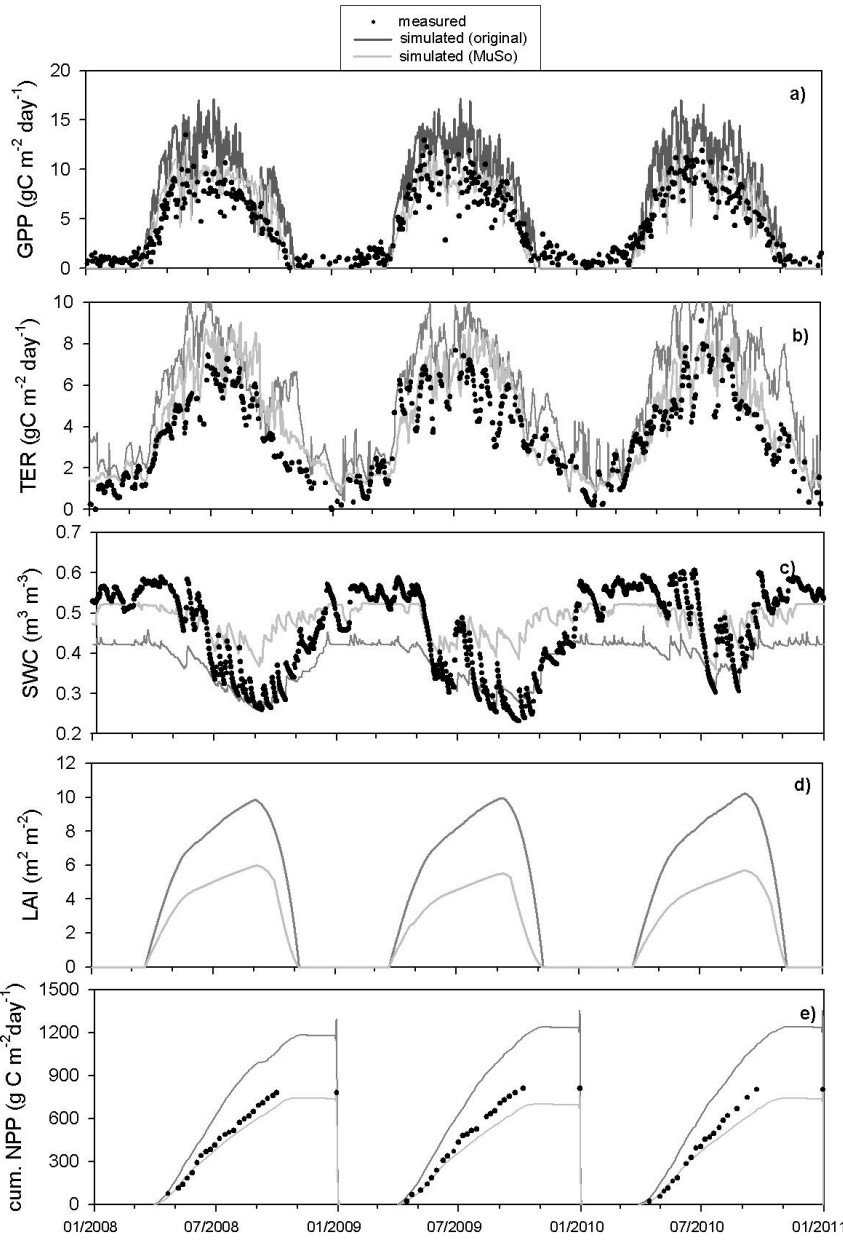

**Figure 8. Measured (black dots) and simulated variables using original Biome-BGC (dark grey lines) and BBGCMuSo (light grey lines) regarding deciduous broad-leaved forest at the Jastrebarsko site between 2008 and 2010. a) GPP, b) TER, c) SWC (note: original Biome-BGC provides simulation results only for one layer, namely 'the rooting depth', which is 0-100 cm in this case), d) LAI, e) cumulative NPP. Note that measured LAI is not available.**



1  **Table 1: Summary of model adjustments. Group refers to major processes represented by collection of modules in**
2  **Biome-BGCMuSo, while sub-group refers to specific features.**

| Group | Sub-group | Description |
|---|---|---|
| **Multilayer soil** | Thermodynamics | Two methods to calculate soil temperature layer by layer: logarithmic and the Ritchie (1998) method |
| | Hydrology | Calculation of soil water content and soil properties layer by layer. New processes: percolation, diffusion, runoff, diffusion, and pond water formation. |
| | Root distribution | Calculation of root mass proportion layer by layer based on empirical function after Jarvis (1989) |
| | Nitrogen budget | Instead of uniform distribution, calculation of soil mineral nitrogen content layer by layer |
| | Soil moisture tress index | Soil moisture stress index based on normalized soil water content calculated layer by layer |
| | Senescence | Senescence calculation based on soil moisture stress index and the number of days since stress is present |
| | Decomposition and respiration | Maintenance root respiration flux is calculated layer by layer. Limitation of soil moisture stress index. |
| **Management** | Management modules | Mowing, grazing, harvest, ploughing, fertilization, plating, thinning, and irrigation. 7 different events for each management activity can be defined per year, even in annually varying fashion. |
| | Management related plant mortality | Rate of the belowground decrease to the mortality rate of the aboveground plant material can be set optionally |
| **Other plant related processes** | Phenology | Alternative model based phenology. Calculating start and end of vegetation period using heat sum index for growing season (extension of index in Jolly et al., 2005) |
| | Genetically programmed leaf senescence | Genetically programmed leaf senescence due to the age of the plant tissue |
| | New plant pools | Fruit and soft stem |
| | $C_4$ photosynthesis | Enzyme-driven C4 photosynthesis routine based on the work of Di Vittorio et al. (2010) |
| | Acclimation | Photosynthesis and respiration acclimation of plants, temperature-dependent Q10 (Smith and Dukes, 2012) |
| | LAI-dependent albedo | LAI-dependent albedo estimation based on the method of Ritchie (1998) |
| | Stomatal conductance regulation | $CO_2$ concentration is taken into account in stomatal conductance estimation based on the Franks et al. (2013) |
| **Model run** | Dynamic mortality | Ecophysiological parameter for whole-plant mortality fraction can be defined year by year |
| | Transient run | Optional third model phase to enable smooth transition from the spinup phase to the normal phase |
| | Possibility of land-use-change simulation | Avoiding frequent model crash in case of different ecophysiological parameters set in spinup and normal simulations |
| **Methane and nitrous oxide soil efflux** | Unmanaged soils | Empirical estimation based on Hashimoto et al. (2011) |
| | Grazing and fertilization | Empirical estimation based on IPCC (2006) Tier 1 method |



Table 2. Default values of parameters in BBGCMuSo for different soil types. Soil types are identified based on the
used-defined sand, silt and clay fraction of the soil layers based on the soil triangle method.

|  | SAND | LOAMY SAND | SANDY LOAM | SANDY CLAY | SANDY CLAY LOAM | LOAM | SILT LOAM | SILT | SILTY CLAY LOAM | CLAY LOAM | SILTY CLAY | CLAY |
|---|---|---|---|---|---|---|---|---|---|---|---|---|
| B | 3.45 | 10.45 | 9.22 | 5.26 | 4.02 | 50 | 7.71 | 7.63 | 6.12 | 5.39 | 4.11 | 12.46 |
| $SWC_{sat}$ | 0.4 | 0.52 | 0.515 | 0.44 | 0.49 | 0.51 | 0.505 | 0.5 | 0.46 | 0.48 | 0.42 | 0.525 |
| $SWC_{fc}$ | 0.155 | 0.445 | 0.435 | 0.25 | 0.38 | 0.42 | 0.405 | 0.39 | 0.31 | 0.36 | 0.19 | 0.46 |
| $SWC_{wp}$ | 0.03 | 0.275 | 0.26 | 0.09 | 0.19 | 0.24 | 0.22 | 0.205 | 0.13 | 0.17 | 0.05 | 0.29 |
| BD | 1.60 | 1.4 | 1.42 | 1.56 | 1.5 | 1.44 | 1.46 | 1.48 | 1.54 | 1.52 | 1.58 | 1.38 |
| RCN | 50 | 70 | 68 | 54 | 56 | 64 | 66 | 62 | 58 | 60 | 52 | 72 |

B [dimensionless]: Clapp-Hornberger parameter; $SWC_{sat}$, $SWC_{fc}$, $SWC_{wp}$ [$m^3$ $m^{-3}$]: SWCs at saturation, at the
field capacity and at the wilting point, respectively; BD [$g$ $cm^{-3}$]: bulk density; RCN [dimensionless]: runoff
curve number.



**Table 3. Quantitative evaluation of the original and adjusted models at the grassland site in Bugac using different**
**error metrics. ORIG means the original Biome-BGC, while MuSo refers the BBGCMuSo. See text for the definition of**
**the error metrics.**

|  | $R^2$ | | RMSE | | NRMSE | | NSE | | BIAS | |
|---|---|---|---|---|---|---|---|---|---|---|
|  | ORIG | MuSo | ORIG | MuSo | ORIG | MuSo | ORIG | MuSo | ORIG | MuSo |
| **GPP** | 0.50 | 0.66 | 5.82 | 1.52 | 51.2 | 13.4 | -4.78 | 0.61 | 2.81 | 0.11 |
| **TER** | 0.65 | 0.64 | 3.21 | 0.98 | 38.1 | 11.6 | -3.59 | 0.57 | 2.47 | 0.33 |
| **LHF** | 0.24 | 0.62 | 1.89 | 0.83 | 33.2 | 14.5 | -1.51 | 0.52 | 0.01 | -0.33 |
| **SWC** | 0.60[*] | 0.64 | 0.04[*] | 0.09 | 34.6[*] | 87.7 | -0.46[*] | -8.3 | -0.01[*] | 0.09 |

[*] SWC simulated by the original Biome-BGC represents constant value within the entire root zone





**Table 4: Quantitative model evaluation regarding to maize simulation at the Mead1 site using different error metrics.**
**ORIG means the original Biome-BGC, while MuSo refers to the BBGCMuSo. See text for the definition on the error**
**metrics.**

| | $R^2$ | | RMSE | | NRMSE | | NSE | | BIAS | |
|---|---|---|---|---|---|---|---|---|---|---|
| | **ORIG** | **MuSo** | **ORIG** | **MuSo** | **ORIG** | **MuSo** | **ORIG** | **MuSo** | **ORIG** | **MuSo** |
| **GPP** | 0.61 | 0.87 | 5.83 | 3.44 | 19.0 | 11.1 | 0.58 | 0.86 | -0.41 | -0.91 |
| **TER** | 0.59 | 0.86 | 4.65 | 1.75 | 35.4 | 13.3 | -0.58 | 0.78 | 3.8 | -0.83 |
| **SWC** | 0.12* | 0.01 | 0.13* | 0.09 | 41.2* | 30.1 | -4.53* | -1.94 | 0.12* | 0.07 |
| **LAI** | 0.26 | 0.59 | 5.81 | 1.61 | 97.6 | 27.1 | -7.03 | 0.91 | 4.22 | -0.01 |
| **agbC** | 0.36 | 0.82 | 296.5 | 159.6 | 27.4 | 14.8 | 0.17 | 0.76 | -129.1 | 7.9 |

* SWC simulated by the original Biome-BGC represents constant value within the entire root zone



**Table 5: Quantitative model evaluation of simulation for deciduous broad-leaved forest for the Jastrebarsko forest**
**site using different error metrics. ORIG means the original Biome-BGC, while MuSo refers to the BBGCMuSo. See**
**text for the definition on the error metrics.**

|  | $R^2$ | | RMSE | | NRMSE | | NSE | | BIAS | |
|---|---|---|---|---|---|---|---|---|---|---|
|  | ORIG | MuSo | ORIG | MuSo | ORIG | MuSo | ORIG | MuSo | ORIG | MuSo |
| **GPP** | 0.84 | 0.84 | 3.67 | 1.77 | 27.3 | 13.1 | -0.13 | 0.74 | 2.27 | 0.33 |
| **TER** | 0.81 | 0.83 | 2.5 | 1.31 | 27.4 | 14.4 | -0.51 | 0.59 | 2.13 | 0.82 |
| **SWC** | 0.83* | 0.67 | 0.11* | 0.08 | 27.1* | 21.2 | 0.06* | 0.42 | -0.08* | 0.02 |
| **NPP** | 0.99 | 0.99 | 240.9 | 70.3 | 30.5 | 8.9 | -0.03 | 0.91 | 220.3 | -52.3 |

* SWC simulated by the original Biome-BGC represents constant value within the entire root zone