# Peer review of "Terrestrial Ecosystem Process Model Biome-BGCMuSo v4.0: 1"

_Geoscientific Model Development, 2016_

## Short Comment (SC1) · 2 Jun 2016

We appreciate the time and efforts by the Editor checking our manuscript. We have addressed all issues indicated in the similarity report, and hope that the revised version may meet the publication requirements. Most of the similarities were due to the manual written by the main authors which is available online. Remaining 7% similarity was caused by similarity of site and model descriptions with our earlier published work. We rephrased these parts in order to avoid copying sentences from other papers even so every similar sentence was written by the authors.

---

## Short Comment (SC2) · 2 Jun 2016

We appreciate the time and efforts by the Editor checking our manuscript. We have addressed all issues indicated in the similarity report, and hope that the revised version may meet the publication requirements. Most of the similarities were due to the manual written by the main authors which is available online. Remaining 7% similarity was caused by similarity of site and model descriptions with our earlier published work. We rephrased these parts in order to avoid copying sentences from other papers even so every similar sentence was written by the authors.

Please also note the supplement to this comment:

http://www.geosci-model-dev-discuss.net/gmd-2016-93/gmd-2016-93-SC2-supplement.pdf

**Supplement:**

[revised manuscript text omitted]

---

## Short Comment (SC3) · 6 Jun 2016

Dear authors,

In my role as Executive editor of GMD, I would like to bring to your attention our Editorial version 1.1:

http://www.geosci-model-dev.net/8/3487/2015/gmd-8-3487-2015.html

This highlights some requirements of papers published in GMD, which is also available on the GMD website in the 'Manuscript Types' section:

http://www.geoscientific-model-development.net/submission/manuscript_types.html

In particular, please note that for your paper, the following requirements have not been met in the Discussions paper:

- "The main paper must give the model name and version number (or other unique identifier) in the title."

- "If the model development relates to a single model then the model name and the version number must be included in the title of the paper. If the main intention of an article is to make a general (i.e. model independent) statement about the usefulness of a new development, but the usefulness is shown with the help of one specific model, the model name and version number must be stated in the title. The title could have a form such as, "Title outlining amazing generic advance: a case study with Model XXX (version Y)"."

Please correct this in your revised submission to GMD.

Yours,

Astrid Kerkweg

---

## Short Comment (SC4) · 8 Jun 2016

Dear Dr. Astrid Kerkweg,

Thank you very much for the note on our manuscript that is under review in Geoscientific Model Development Discussions (gmd-2016-93). We appreciate your time and efforts to check our manuscript. The manuscript describes a single model, Biome-BGCMuSo v4.0. According to the note (gmd-2016-93-SC3) we are going to extend the title of our manuscript so that will include the version number of the model in the revised manuscript.

Yours sincerely, Dóra HIDY and Zoltán BARCZA

---

## Referee Comment (RC1) · Anonymous Referee #1 · 9 Jun 2016

Comments to "Hidy et al., Terrestrial Ecosystem Process Model Biome-BGCMuSo: Summary of improvements and new modelling possibilities".

Overall: This manuscript introduced new developments of the widely-used terrestrial ecosystem model Biome-BGC, particularly focusing on soil dynamics. The modified Biome-BGC (named as Biome-BGCMuSo) incorporated several new modules that were non-existed in the original Biome-BGC, such as multiple soil layers, management, plant physiological processes, and other GHG simulation modules.

I admire the great effort spent on new developments of Biome-BGC. As the authors indicated, the model still has been widely used even nowadays, but unfortunately its development halted for some reason. I really encourage continuing the development

further; however, I have to say that there is a serious issue in the modified model.

A major problem is that the authors left the issue of negative nitrogen pool unresolved, in favor of the management module (P27 L3-12). This is a red flag in ecosystem modelling! If the authors are anticipated to employ the MuSo as a part of ESM in near future (P3 L2-13) , this issue definitely need to be resolved with a more proper approach. Otherwise, the future projection of nitrogen cycle will be biased by non-equilibrium state.

Most importantly, with the current version of MuSo, the authors cannot claim legitimacy of carbon cycle as well. Biome-BGC is a carbon/nitrogen coupled model. Therefore, distorted nitrogen cycle affects carbon cycle, especially in long-term simulations.

The manuscript provided many good pieces of information about the validation of the modified model, but they are rather meaningless without a proper treatment of carbon/nitrogen cycles. So I don't comment on them at this stage.

The full modification of the model would take time, so I recommend to withdraw the current manuscript and resubmit a new one later.

---

## Referee Comment (RC2) · B. Bond-Lamberty (Referee) · 9 Jun 2016

General comments

—————————-

This mansuscript describes and documents a new version of the classic and widely-used ecosystem model Biome-BGC. The new version (Biome-BGCMuSo) has a wide variety of new physiological mechanisms, management-simulating routines, soil processes, etc. A number of case studies are described, in which the authors run the new model and assess its performance at a variety of sites. In general everything is well described, and basically clear.

There are a few weaknesses, only one of which I consider critical. First, the processes that have been implemented or improved come across as something of a laundry list, and a bit random; it would be good to describe any common themes that unite them, and better talk about remaining areas of weakness the authors see in the model.

Second, there's no validation or testing of the individual changes. This isn't a show-stopper, but it would have been really useful to run many more simulations with and without each of the new changes. That would have let you quantitatively assess their individual value.

Finally, and most seriously, I'm mystified that the authors are choosing to host the code on their own webserver. GitHub has become the standard for scientific software repositories, but even if you don't use it, for permanence and reproducibility you HAVE to use version control (so, for example, people can send you pull requests or see commit histories) in an established repository. This would massively increase the utility of the new model, and enable it to become a true community-driven model (or, at least, a much more transparent one). I applaud all the work you've done here, including on this website, but really, really urge you to make use of GitHub or another web-based repository hosting service. Note that I have posted the Biome-BGC source code in such a repository (https://github.com/bpbond/Biome-BGC) and would be happy to make this available, and/or turn it over to you.

Specific comments

————————————-

1. Page 3, line 4: probably "often is no longer" (concentration-driven runs remain very common in ESM simulations)

2. P. 3, l. 8-9: perhaps cite one of the Friedlingstein et al. papers (e.g. 10.1175/JCLI-D-12-00579.1)

3. P. 3, l. 30: perhaps cite Thornton et al. (2013) 10.1029/2006GB002868

4. P. 4: wow, great summary of Biome-BGC history! One addition might be in line 12: ". . .and decomposition, and then simulated wildfire effects across a western Canadian forest landscape (Bond-Lamberty et al. 2007, 10.1038/nature06272)." (Completely optional though.)

5. P. 5, l. 11-12: I think this is true, but would add that Biome-BGC also strikes a great balance between process fidelity and tractability: it's relatively easy to use and run, even for non-specialists, but still yields interesting insights. If you agree, perhaps note this

6. P. 10, l. 21-22: More dials and knobs aren't always better. How do you know this is an improvement? Are four layers known to be too few?

7. P. 12, l. 36-37: a thought for the future: an adaptive-timestep algorithm that shrinks the timestep only when necessary

8. P. 27, l. 3-12: nice

9. P. 31, l. 19: "The probable"

10. P. 31, l. 35: "The original"

11. P. 32, l. 32: "The role"

12. P. 33, l. 34-: consider mentioning/discussing PEcAn – see Dietze et al. (2014) 10.1002/2013JG002392

---

## Short Comment (SC5) · 15 Jun 2016

Reply to the interactive comment "RC1: 'Improper compensation between nitrogen cycle and management module', by Anonymous Referee 1, 09 Jun 2016" by Dóra HIDY and Zoltán BARCZA

We thank the Anonymous Referee for the valuable comments on our manuscript (gmd-2016-93). The notes of the Referee are inserted below in between quotation marks in italic. Our detailed response to the comments and suggestions are presented below.

[Figure]

*"A major problem is that the authors left the issue of negative nitrogen pool unresolved, in favor of the management module (P27 L3-12). This is a red flag in ecosystem modelling! If the authors are anticipated to employ the MuSo as a part of ESM in near future (P3 L2-13), this issue definitely need to be resolved with a more proper approach. Otherwise, the future projection of nitrogen cycle will be biased by non-equilibrium state."*

We believe that this is a misunderstanding. In Biome-BGCMuSo v4.0, and also during the development of earlier Biome-BGCMuSo versions, **we have fixed the issue of negative nitrogen (N) pool**. In other words, the negative nitrogen pool issue is completely resolved in the model version that we describe in the manuscript. Negative nitrogen pool **was** an issue in earlier model versions in some cases during land use change simulations (LUC; when normal phase represent a different plant functional type than the spinup phase) that we mention in the original manuscript (P27 L3-12). We realized that the original text could be misunderstood, as we wrote the following: "One problem that is associated with LUC is the frequent crash of the model with the error 'negative nitrogen pool' during the beginning of the normal phase." It would have been better to write it as "....One problem that was associated with LUC in earlier model versions was the frequent...". In the revised manuscript we will rephrase this sentence to avoid misunderstanding.

Below we provide a brief description of the logic that is used in the LUC simulations for clarity. This implementation can be traced by checking the model code which is available at the model's website at

http://nimbus.elte.hu/bbgc/files/biome-bgc-muso_4.0_2016_03_03_Linux_src.tar

The core logic of all Biome-BGC versions is that the N flux calculations of the plant compartments are based on the carbon fluxes of the corresponding plant compartments and the C:N ratios of the plant compartments. The logic for the carbon fluxes is different because they are based on the content of the carbon pools and the environmental drivers. If the C:N ratio of a given plant compartment in the normal phase

is lower than in the spinup phase, it can happen that the N flux calculated from the C flux using the new C:N ratio is higher than the content of the N pool. In earlier model versions this caused negative N pool thus crashing of the simulation. In order to avoid this error we have implemented an automatic procedure (in the source code this is implemented in restart_io.c): in the normal phase, instead of using the result of the spinup phase, the nitrogen content of the plant compartments are calculated internally by the model code based on the final (equilibrium) carbon pools and the new C:N ratios of the plant compartments, so that the resulting N pools of the plant compartments are harmonized with the C:N ratio of the plant compartments presented in the ecophysiological parameterization.

It is important to note that soil and litter pools (litter labile, unshielded cellulose, shielded cellulose and lignin C and N) are not affected by this LUC handler, only the actual, transfer and storage pools of leaf, fine root, soft stem, fruit, liveroot, deadroot, livestem and deadstem. It means that equilibrium C and N soil pools (which are the most important elements for the initial conditions of the normal run) remain intact. In the manuscript we stated that the equilibrium N pools are adjusted, but the word "plant" was missing, thus it was ambiguous. In the revised manuscript we will correct this problem ("equilibrium carbon pools" in P27 L8 will be replaced with "equilibrium plant carbon pools", and the same for nitrogen).

In case of the Mead simulation (presented in our manuscript) where the spinup phase used C3grass parameterization and the normal phase used maize parameterization this modification resulted 0.49 g N m-2 N-surplus in transfer N pool of fine root (because the fine root C:N ratio of maize was lower than the fine root C:N ratio of C3grass). The magnitude of the change is similar in case of other LUC types such as forest-herbaceous, herbaceous-forest, and herbaceous-herbaceous. But most importantly **in all possible LUC cases the plant N pools that are adjusted are not important anymore, since they are replaced by new vegetation in the normal phase with new C:N stoichiometry**. We added a new sentence to the revised manuscript to

explain this logic: "This modification means that equilibrium plant nitrogen pools are not passed to the normal phase, but in fact this is not needed because LUC means change in the existing plant functional type, so new plant C:N ratios will inevitably be realized."

All in all, the equilibrium soil carbon and nitrogen pools are not changed during LUC, only the plant compartments that are in fact replaced by a new plant functional type during the first year of the normal run. Most importantly, this handler resolves the negative nitrogen pool issue that was possible in earlier model versions. We truly believe that this is a remarkable improvement as previous, published and widely used Biome-BGC model versions frequently crashed in such situations.

*"Most importantly, with the current version of MuSo, the authors cannot claim legitimacy of carbon cycle as well. Biome-BGC is a carbon/nitrogen coupled model. Therefore, distorted nitrogen cycle affects carbon cycle, especially in long-term simulations. The manuscript provided many good pieces of information about the validation of the modified model, but they are rather meaningless without a proper treatment of carbon/nitrogen cycles. So I don't comment on them at this stage."*

We kindly ask the Referee to comment on the validation of the model (if he/she accepts our explanation on the negative nitrogen pool). We appreciate any comments about the case studies and about the rest of the manuscript.

*"The full modification of the model would take time, so I recommend to withdraw the current manuscript and resubmit a new one later."*

We hope that the Referee agrees that it is not necessary to withdraw the manuscript as the negative nitrogen pool issue does not exist in Biome-BGCMuSo v4.0.

---

## Author Comment (AC2) · 14 Jul 2016

Dear Editor, Dear Reviewers,

The attached revised manuscript contains all modifications that were made according the the comments of Anomynous Reviewer 1 (RC1) and Reviewer 2 (RC2, Ben Bond-Lamberty). We also corrected the manuscript according to the Similarity check report that was made on the original submission. We also added indication of model version number according to the note of Dr. Astrid Kerkweg.

Yours sincerely, Dora HIDY and Zoltan BARCZA

Please also note the supplement to this comment:
http://www.geosci-model-dev-discuss.net/gmd-2016-93/gmd-2016-93-AC2-
supplement.pdf

**Supplement:**

[revised manuscript text omitted]
} \mathrm{NSWC} &= \frac{\mathrm{SWC} - \mathrm{SWC_{wp}}}{\mathrm{SWC_{sat}} - \mathrm{SWC_{wp}}}, \text{if } \mathrm{SWC_{wp}} < \mathrm{SWC} \\ \mathrm{NSWC} &= 0 \qquad\qquad\quad\ \ , \text{if } \mathrm{SWC_{wp}} \geq \mathrm{SWC} \end{aligned} \qquad (6)$$

where SWC is the soil water content of the given soil layer ($m^3$ $m^{-3}$), $\mathrm{SWC_{wp}}$ and $\mathrm{SWC_{sat}}$ are the wilting point and the saturation value of soil water content, respectively. Parameters can either be set by the user or calculated by the model internally (see Table 2).

The NSWC and the soil water potential as function of SWC are presented in Fig. 1 for three different soil types:

sand soil (SAND: 90%, SILT: 5%), sandy clay loam (SAND: 50%, SILT: 20%), and clay soil (SAND: 8%,

SILT: 45%).

According to our definition SMSI is a function of NSWC and can vary between 0 (maximum stress) and 1

(minimum stress).

The general form of SMSI is defined by the following equations:

$$\begin{aligned} \mathrm{SMSI} &= \frac{\mathrm{NSWC}}{\mathrm{NSWC_{crit1}}}, \quad \text{if } \mathrm{NSWC} < \mathrm{NSWC_{crit1}} \\ \mathrm{SMSI} &= 1, \qquad\qquad\quad \text{if } \mathrm{NSWC_{crit1}} < \mathrm{NSWC} \leq \mathrm{NSWC_{crit2}} \\ \mathrm{SMSI} &= \frac{1 - \mathrm{NSWC}}{1 - \mathrm{NSWC_{crit2}}}, \text{if } \mathrm{NSWC_{crit2}} < \mathrm{NSWC} \end{aligned} \qquad (7)$$

[revised manuscript text omitted]

---

## Author Response (AR1)

[revised manuscript text omitted]

Geosci. Model Dev. Discuss.,
doi:10.5194/gmd-2016-93-AC1, 2016

[Figure]

Reply to the interactive comment "RC2: 'Valuable step and contribution weakened by no repository', by Ben Bond-Lamberty (Referee 2), 09 Jun 2016" by Zoltán BARCZA and Dóra HIDY

First of all, we thank Ben Bond-Lamberty for reviewing the manuscript (gmd-2016-93), we are grateful for the positive and valuable notes. The comments of the Referee are shown below in between quotation marks in italic. Our detailed response to the comments and suggestions are presented below. We attached the revised manuscript to this interactive comment. The revised manuscript also contains modifications to fix

the issues raised by the Anonymous Referee #1.

*"There are a few weaknesses, only one of which I consider critical. First, the processes that have been implemented or improved come across as something of a laundry list, and a bit random; it would be good to describe any common themes that unite them, and better talk about remaining areas of weakness the authors see in the model."*

We agree with the Referee that the improvements might seem a bit random. In fact we are working on the developments for quite a long time, and during the years we faced different problems with the model, and we tried to fix them using our best knowledge. Three common themes emerge (two larger groups plus a group of heterogeneous fixes): soil processes (from different aspects), management (now it covers the majority of typical human interventions), and a number of additional fixes and adjustments to create a state-of-the are Biome-BGC (we can mention here acclimation, representation of the $CO_2$ mixing ratio dependence of stomatal conductance, and estimation of other greenhouse gas fluxes). We truly hope that this is not a serious problem as the modeling community might find these diverse developments useful in many cases. Publication of partial adjustments is an option, but we decided to disseminate the developments as part of this major work for easier accessibility and reference.

Nevertheless, we added a few sentences to the revised manuscript (P9 L24-38) where we mention the main motivations for the adjustments: "The developments were motivated by multiple factors. Poor agreement of the modified Biome-BGC with available eddy covariance measurements made in Hungary over two grassland sites (see Hidy et al., 2012) clearly revealed the need for more sophisticated representation of soil hydrology, especially at the drought-prone, sandy Bugac site. Implementation and benchmarking of the multilayer soil module resulted in several additional developments related e.g. to the N balance, soil temperature, root profile, soil water deficit effect on plant functioning and decomposition of soil organic matter. Lack of management descriptor within the original Biome-BGC motivated the development of a broad array of possible management techniques covering typical grassland, cropland and forest

management practices. Modeling exercises within international projects revealed additional problems that needed a solution related e.g. to stomatal conductance (Sándor et al., 2016). Recently published findings (e.g. simulation of temperature acclimation of respiration; Smith and Dukes, 2012) also motivated our work on model development. Addressing the known issues with the Biome-BGC model required diverse directions but this was necessary due to the complex nature of the biogeochemical cycles of the soil-plant system. Our overarching aim was to provide a state-of-the-art biogeochemical model that is in the same league as currently available models, such as LPJmL, ORCHIDEE, CLM, JULES, CASA and others."

In the 'Discussion and conclusions' section we mention a few remaining areas where developments are needed in the future:

"Representation of photosynthesis acclimation (Medlyn et al., 2002) is needed in future modifications as another major development." (P32 L29-31 in the revised MS)

"Developments are clearly needed in terms of soil water balance and ecosystem scale hydrology in general. Other developments are still needed to improve simulations with dynamic C and N allocation within the plant compartments (Friedlingstein et al., 1999; Olin et al., 2015). Complete representation of ammonium and nitrate pools with associated nitrification and denitrification is also needed to avoid ill-defined, N balance related parameters (Thomas et al., 2013). Even further model development will require the addition of other nutrient limitations (phosphorus, potassium)." (P33 L3-8 in the revised MS)

"Clearly, extensive testing is required at multiple EC sites to evaluate the performance of the model and to make adjustments if needed." (P33 L10-11 in the revised MS)

*"Second, there's no validation or testing of the individual changes. This isn't a showstopper, but it would have been really useful to run many more simulations with and without each of the new changes. That would have let you quantitatively assess their*

[Figure]

*individual value."*

We presented several additional simulation results in the Supplementary material (S2-S9): http://www.geosci-model-dev-discuss.net/gmd-2016-93/gmd-2016-93-supplement.pdf

In the Supplementary material we illustrate the effect of different model features such as senescence, management, and groundwater effects on the simulated results in a fashion that is mentioned by the Referee (switching on and off individual features to see its result on the simulated fluxes and pools). We hope that this is what the Referee proposed.

*"Finally, and most seriously, I'm mystified that the authors are choosing to host the code on their own webserver. GitHub has become the standard for scientific software repositories, but even if you don't use it, for permanence and reproducibility you HAVE to use version control (so, for example, people can send you pull requests or see commit histories) in an established repository. This would massively increase the utility of the new model, and enable it to become a true community-driven model (or, at least, a much more transparent one). I applaud all the work you've done here, including on this website, but really, really urge you to make use of GitHub or another web-based repository hosting service. Note that I have posted the Biome-BGC source code in such a repository (https://github.com/bpbond/Biome-BGC) and would be happy to make this available, and/or turn it over to you."*

Thank you very much for this proposal. We did not use GitHub (or other source code repository) or other version control software previously. We have submitted the source code to GitHub, which is available online at https://github.com/bpbond/Biome-BGC/tree/Biome-BGCMuSo_v4.0

We have inserted a sentence into the manuscript into Code availability section about this possiblity (P34 L5-6 in the revised manuscript).

[Figure]

*Specific comments*

*"1. Page 3, line 4: probably 'often is no longer' (concentration-driven runs remain very common in ESM simulations)"*

We have modified the sentence according to the Referee's comment (P3 L3 in the revised manuscript).

*"2. P. 3, l. 8-9: perhaps cite one of the Friedlingstein et al. papers (e.g. 10.1175/JCLI-D-12-00579.1)"*

We have inserted the Friedlingstein et al. citation.

*"3. P. 3, l. 30: perhaps cite Thornton et al. (2013) 10.1029/2006GB002868"*

We have inserted the Thornton et al. citation.

*"4. P. 4: wow, great summary of Biome-BGC history! One addition might be in line 12: '. . .and decomposition, and then simulated wildfire effects across a western Canadian forest landscape (Bond-Lamberty et al. 2007, 10.1038/nature06272).' (Completely optional though.)"*

Thank you for mentioning this paper. Somehow we overlooked this important paper. We have inserted the text and the citation that the Referee suggested into the revised manuscript.

*"5. P. 5, l. 11-12: I think this is true, but would add that Biome-BGC also strikes a great balance between process fidelity and tractability: it's relatively easy to use and run, even for non-specialists, but still yields interesting insights. If you agree, perhaps note this"*

We have inserted this note to the manuscript. In the revised MS (P5 L11-14) the sentence reads as: "The success and widespread application of Biome-BGC can be

attributed to the fact that Biome-BGC strikes a great balance between process fidelity and tractability: it is relatively easy to use and run, even for non-specialists, but still yields interesting insights. The success can also be attributed to the open source nature of the model code."

*"6. P. 10, l. 21-22: More dials and knobs aren't always better. How do you know this is an improvement? Are four layers known to be too few?"*

The motivation for the implementation of 7 soil layers can be summarized with the followings: - Using 7 layers the soil profile can be better described, which means that change of texture and bulk density with depth can be described in a more realistic fashion. Soil profile effect on soil hydrology was demonstrated in many cases (this feature is also used for the Jastrebarsko case study within the paper), and it can be important in many cases. - Implementation of thin soil layers close to the surface can support drying out the upper layers (due to surface evaporation and root water uptake which is strong close to the surface), so the resulting soil water content profile is closer to observations. Consequently, drought effect on plant functioning and soil microbial activity is better captured. - Usage of multiple layers (with thinner layers close to the soil surface) might be important for herbaceous vegetation due to the small rooting depth. - Cropland related developments have priority in the future within Hungarian research projects, and experience from crop models (e.g. from DSSAT) shows the necessity of a larger number of simulated soil layers.

In summary, we think that 4 soil layers are too few (even 7 layers might turn out to be too few) especially in case of herbaceous vegetation. Benchmarking and further development of the model is needed to find the optimal distribution of soil layers.

We have extended the manuscript summarizing the above reasoning (P10 L22-28 in the revised MS): "Higher number of soil layers might be useful to better represent the soil profile in terms of soil texture and bulk density. This should be beneficial for sites with complex hydrology (see e.g. the Jastrebarsko case study in section 5.3).

[Figure]

Additional layers also improve the representation of soil water content profile as the upper, thinner soil layers typically dry out more than the deeper layers, which affects soil and plant processes. As rooting depth can be quite variable, additional layers might support the proper representation of soil water stress on plant functioning. Seven layers provide an optimal compromise between simulation accuracy and computational cost."

*"7. P. 12, l. 36-37: a thought for the future: an adaptive-timestep algorithm that shrinks the timestep only when necessary"*

Thank you for the note. In fact we implemented such solution that is suggested by the Referee. P13 L1-15 in the original manuscript described this feature.

*"9. P. 31, l. 19: 'he probable'"*

Thank you for the language corrections. We modified the sentence.

*"10. P. 31, l. 35: 'The original'"*

We have corrected this.

*"11. P. 32, l. 32: 'The role'"*

We made the correction.

*"12. P. 33, l. 34-: consider mentioning/discussing PEcAn ' see Dietze et al. (2014) 10.1002/2013JG002392"*

Thank you for mentioning PEcAn, this workflow based system is indeed similar to the one we implemented. We have inserted a sentence to the revised manuscript (P33 L24-26) mentioning PEcAn and the ED model: "The infrastructure is similar to PEcAn (Dietze et al., 2014) which is a collection of modules in a workflow that uses the Ecosystem Demography (ED v2.2) model."

Please also note the supplement to this comment:
http://www.geosci-model-dev-discuss.net/gmd-2016-93/gmd-2016-93-AC1-

supplement.pdf

[Figure]

Geosci. Model Dev. Discuss.,
doi:10.5194/gmd-2016-93-SC5, 2016

[Figure]

Reply to the interactive comment "RC1: 'Improper compensation between nitrogen cycle and management module', by Anonymous Referee 1, 09 Jun 2016" by Dóra HIDY and Zoltán BARCZA

We thank the Anonymous Referee for the valuable comments on our manuscript (gmd-2016-93). The notes of the Referee are inserted below in between quotation marks in italic. Our detailed response to the comments and suggestions are presented below.

[Figure]

*"A major problem is that the authors left the issue of negative nitrogen pool unresolved, in favor of the management module (P27 L3-12). This is a red flag in ecosystem modelling! If the authors are anticipated to employ the MuSo as a part of ESM in near future (P3 L2-13), this issue definitely need to be resolved with a more proper approach. Otherwise, the future projection of nitrogen cycle will be biased by non-equilibrium state."*

We believe that this is a misunderstanding. In Biome-BGCMuSo v4.0, and also during the development of earlier Biome-BGCMuSo versions, **we have fixed the issue of negative nitrogen (N) pool**. In other words, the negative nitrogen pool issue is completely resolved in the model version that we describe in the manuscript. Negative nitrogen pool **was** an issue in earlier model versions in some cases during land use change simulations (LUC; when normal phase represent a different plant functional type than the spinup phase) that we mention in the original manuscript (P27 L3-12). We realized that the original text could be misunderstood, as we wrote the following: "One problem that is associated with LUC is the frequent crash of the model with the error 'negative nitrogen pool' during the beginning of the normal phase." It would have been better to write it as "....One problem that was associated with LUC in earlier model versions was the frequent...". In the revised manuscript we will rephrase this sentence to avoid misunderstanding.

Below we provide a brief description of the logic that is used in the LUC simulations for clarity. This implementation can be traced by checking the model code which is available at the model's website at http://nimbus.elte.hu/bbgc/files/biome-bgc-muso_4.0_2016_03_03_Linux_src.tar

The core logic of all Biome-BGC versions is that the N flux calculations of the plant compartments are based on the carbon fluxes of the corresponding plant compartments and the C:N ratios of the plant compartments. The logic for the carbon fluxes is different because they are based on the content of the carbon pools and the environmental drivers. If the C:N ratio of a given plant compartment in the normal phase

is lower than in the spinup phase, it can happen that the N flux calculated from the C flux using the new C:N ratio is higher than the content of the N pool. In earlier model versions this caused negative N pool thus crashing of the simulation. In order to avoid this error we have implemented an automatic procedure (in the source code this is implemented in restart_io.c): in the normal phase, instead of using the result of the spinup phase, the nitrogen content of the plant compartments are calculated internally by the model code based on the final (equilibrium) carbon pools and the new C:N ratios of the plant compartments, so that the resulting N pools of the plant compartments are harmonized with the C:N ratio of the plant compartments presented in the ecophysio-logical parameterization.

It is important to note that soil and litter pools (litter labile, unshielded cellulose, shielded cellulose and lignin C and N) are not affected by this LUC handler, only the actual, transfer and storage pools of leaf, fine root, soft stem, fruit, liveroot, deadroot, livestem and deadstem. It means that equilibrium C and N soil pools (which are the most important elements for the initial conditions of the normal run) remain intact. In the manuscript we stated that the equilibrium N pools are adjusted, but the word "plant" was missing, thus it was ambiguous. In the revised manuscript we will correct this problem ("equilibrium carbon pools" in P27 L8 will be replaced with "equilibrium plant carbon pools", and the same for nitrogen).

In case of the Mead simulation (presented in our manuscript) where the spinup phase used C3grass parameterization and the normal phase used maize parameterization this modification resulted 0.49 g N m-2 N-surplus in transfer N pool of fine root (because the fine root C:N ratio of maize was lower than the fine root C:N ratio of C3grass). The magnitude of the change is similar in case of other LUC types such as forest-herbaceous, herbaceous-forest, and herbaceous-herbaceous. But most importantly **in all possible LUC cases the plant N pools that are adjusted are not important anymore, since they are replaced by new vegetation in the normal phase with new C:N stoichiometry**. We added a new sentence to the revised manuscript to

explain this logic: "This modification means that equilibrium plant nitrogen pools are not passed to the normal phase, but in fact this is not needed because LUC means change in the existing plant functional type, so new plant C:N ratios will inevitably be realized."

All in all, the equilibrium soil carbon and nitrogen pools are not changed during LUC, only the plant compartments that are in fact replaced by a new plant functional type during the first year of the normal run. Most importantly, this handler resolves the negative nitrogen pool issue that was possible in earlier model versions. We truly believe that this is a remarkable improvement as previous, published and widely used Biome-BGC model versions frequently crashed in such situations.

*"Most importantly, with the current version of MuSo, the authors cannot claim legitimacy of carbon cycle as well. Biome-BGC is a carbon/nitrogen coupled model. Therefore, distorted nitrogen cycle affects carbon cycle, especially in long-term simulations. The manuscript provided many good pieces of information about the validation of the modified model, but they are rather meaningless without a proper treatment of carbon/nitrogen cycles. So I don't comment on them at this stage."*

We kindly ask the Referee to comment on the validation of the model (if he/she accepts our explanation on the negative nitrogen pool). We appreciate any comments about the case studies and about the rest of the manuscript.

*"The full modification of the model would take time, so I recommend to withdraw the current manuscript and resubmit a new one later."*

We hope that the Referee agrees that it is not necessary to withdraw the manuscript as the negative nitrogen pool issue does not exist in Biome-BGCMuSo v4.0.

---

## Referee Report (RR1)

Thorough descriptions, but ambiguities in validations

Comments to "Hidy et al., Terrestrial Ecosystem Process Model Biome-BGCMuSo v4.0: Summary of improvements and new modelling possibilities".

Overall:

Thank you for clarifying the nitrogen balance issue, and sorry for misunderstanding. The solution that the authors chose for the negative nitrogen balance issue may not the best, but it is acceptable as a temporal one. I think that the authors will find a better solution eventually in the future.

Now let me comments on validations:

(1) The authors presented comparison between the MuSo and original Biome-BGC with the selected variables (i.e., GPP, TER, SWC, LAI, LHF, and abgC, cumulative NPP) for three different vegetation types (Figure 6-8), and they claimed improvements of the MuSo over the original. However, it is difficult to get a grasp of how/which the new modules contributed to the improvements from the context. It is pity that the authors chose to show some of explicit effects of the new modules in supplemental materials, not in the manuscript. Personally, I like to recommend a thorough reorganization of case studies in such a way to demonstrate more explicit effects of the new modules in relation to $CO_2$ and $H_2O$ fluxes, and storages, but I leave it to discretion of the authors.

(2) Validations against observed $CO_2$ and $H_2O$ fluxes indicate that the new model has better capability of reproducing fluxes of different vegetation types than the original, but I have concerns about reproducibility of LAI.

The average air temperature at the Bugac site is around 10 ˚C (P5 L30), so I expect that the winter temperatures would be near 0 ˚C or can be even negative ˚C at the site. In such a condition, how come LAI starts increasing in January (Figure 6d)? Although *Festuca* and *Carex* are evergreen species, this phenological pattern is unreasonable. Indeed, the average LAI by MuSo is close to the observed value (L26-27, p30), but it does not mean that the phenological pattern by MuSo is more realistic than that by the original.

It is well known that LAI tends to saturate earlier than net carbon uptake (e.g., abgC), which is well illustrated by observed values in the Mead1 site (Figure 7d, e). It seems that MuSo is incapable of reproducing this pattern, both LAI and agbC shows a similar seasonal variability with peaks in a similar time period.

Because of these results, I'm skeptical about performance of the phenology model, particularly heat sum growing index (section 4.3.1). There is no need to modify model results, but instead I insist on describing these results as a limitation. They should be in a list for future modifications (P33 L18-22).

---

## Author Response (AR2)

**Reply to the review**

*"Comments to "Hidy et al., Terrestrial Ecosystem Process Model Biome-BGCMuSo v4.0: Summary of improvements and new modelling possibilities" by Anonymous Referee #1*

First of all, we thank the Referee for reviewing the manuscript again. We are grateful for the positive and valuable notes and we appreciate the time and efforts of the Referee.
The comments of the Referee are shown below between quotation marks in *italic*. Our detailed response is presented below.

*"Thank you for clarifying the nitrogen balance issue, and sorry for misunderstanding. The solution that the authors chose for the negative nitrogen balance issue may not the best, but it is acceptable as a temporal one. I think that the authors will find a better solution eventually in the future."*

We are happy that we succeeded in resolving the misunderstanding. In the forthcoming model version we'll try to find a better solution to handle land use change.

*"The authors presented comparison between the MuSo and original Biome-BGC with the selected variables (i.e., GPP, TER, SWC, LAI, LHF, and abgC, cumulative NPP) for three different vegetation types (Figure 6-8), and they claimed improvements of the MuSo over the original. However, it is difficult to get a grasp of how/which the new modules contributed to the improvements from the context. It is pity that the authors chose to show some of explicit effects of the new modules in supplemental materials, not in the manuscript. Personally, I like to recommend a thorough reorganization of case studies in such a way to demonstrate more explicit effects of the new modules in relation to CO2 and H2O fluxes, and storages, but I leave it to discretion of the authors."*

We agree with the Referee that due to the high number of changes in the model structure it is difficult to say which modification is responsible for the performance improvement. In fact in this manuscript the main aim was to document the changes, and to demonstrate the usefulness of the changes. It was not meant to attribute performance to the specific changes. In this sense the effect of individual changes (features on/off type experiments) is not the main focus, but rather the demonstration for the usefulness of the new features. Nevertheless we agree that it would be good to show the explicit effects of the new modules by moving case studies from the Supplementary material to the Manuscript. After discussion among the authors we opted for leaving the case studies in the Supplement due to very simple financial reasons: in Geoscientific Model Development the cost of publication is proportional with the length of the paper, and given the current financial state of our group we cannot cover the cost of a longer manuscript (funding for open science is very limited in Hungary and Croatia). However, we believe that with the publication of more detailed results from case studies in the Supplement we have provided the scientific community with the opportunity to evaluate our work and to assess the performance of the Biome-BGCMuSo with respect to Biome-BGC on selected sites. We hope that the Referee will accept our decision. In forthcoming papers the usefulness and explicit effect of the new features will be evaluated in detail.

*"Validations against observed $CO_2$ and $H_2O$ fluxes indicate that the new model has better capability of reproducing fluxes of different vegetation types than the original, but I have*

*concerns about reproducibility of LAI. The average air temperature at the Bugac site is around 10 ˚C (P5 L30), so I expect that the winter temperatures would be near 0 ˚C or can be even negative ˚C at the site. In such a condition, how come LAI starts increasing in January (Figure 6d)? Although Festuca and Carex are evergreen species, this phenological pattern is unreasonable. Indeed, the average LAI by MuSo is close to the observed value (L26-27, p30), but it does not mean that the phenological pattern by MuSo is more realistic than that by the original.*

Following the suggestion of the Referee in the revised manuscript we acknowledge that the phenology submodel requires further adjustments and possible development.
As HSGSI was published earlier (Hidy et al., 2012) it was not discussed extensively in the present manuscript. Therefore it might not be intuitive that in fact HSGSI is not a static routine but it can be parameterized in a very flexible manner. Here, for clarity, we briefly describe the logic of the HSGSI-based routine.
HSGSI (abbreviation of Heatsum Growing Season Index) is calculated based on the combination of daily minimum temperature ($T_{min}$), vapor pressure deficit (VPD), daylength (DL) and 10-days heatsum ($HS_{10}$). $HS_{10}$ is calculated as the sum of differences between mean daily temperatures ($T_{avg}$) minus the base temperature ($T_{basic}$, a parameter that is typically set to a specific constant value, e.g. 5 ºC for grasses) for each 10-days long period (number of the averaging days is also an adjustable parameter but here we use 10-days for simplicity):

$$HS_{10}(d) = \sum_{d-9}^{d} (diff_d)$$

$$diff_d = T_{avg}(d) - T_{basic} ; if \ T_{avg}(d) \geq T_{basic}$$

$$diff_d = 0 \qquad\qquad ; if \ T_{avg}(d) < T_{basic}$$

The effect of the different environmental variables is considered using indices (*index_{Tmin}*, *index_{VPD}*, *index_{DL}*, *index_{HS10}*). To calculate index for each variable we set threshold limits (upper and lower limits: Ulimit, Llimit), similarly to the method proposed by Jolly et al. (2005) within which the relative phenological performance of the vegetation was assumed to vary from inactive (0) to unconstrained (1) using ramp function. The values of the limits of these variables can be set as user defined parameters in Biome-BGCMuSo v4.0.
From a given day of the year HSGSI is calculated from the 10-day average of the multiplication of daily indices and from the index of heatsum:

$$HSGSI(d) = index_{HS10} \cdot \left( \frac{1}{10} \cdot \sum_{d-9}^{d} (index_{Tmin}(d) \cdot index_{VPD}(d) \cdot index_{DL}(d)) \right)$$

where d is the number of day of year.
If, on a given day, HSGSI is greater than a predefined limit (adjustable parameter; $HSGSI_{start}$=0.1 was used here) we assume the start of the growing season was found. After finding the start of the growing season its last day is searched, and if on a given day (after start of the growing season) HSGSI is less than a limit (can be set by the user, which was 0.01 in this run) we assume that the end of the growing season was reached.
In Bugac, VPD and DL do not limit the HSGSI calculation, but instead $T_{min}$ and HS are the important limitation factors. In the present study the following limits were used: $Llimit_{HS10}$=1, $Ulimit_{HS10}$=10, $Llimit_{Tmin}$=-2, $Ulimit_{Tmin}$=5, $Llimit_{VPD}$=4000, $Ulimit_{VPD}$=1000, $Llimit_{DL}$=25000 $Ulimit_{DL}$=30000.
In Bugac, during the study period (2009-2011) all three years started with a warmer period with average temperature above 5 ºC and 10-day heatsum greater than 5-10 ºC/10-day, with the minimum temperature greater than -1 °C. Therefore the HSGSI grew larger than a predefined limit before day of year 40. A quick adjustment of the parameters (Llimit$_{HS10}$ and Ulimit$_{HS10}$ changed to 20 and 30, respectively) would resolve the problem.

But again, we would like to emphasize that in this paper adjustment (optimization) of the parameters was not pursued, which resulted in this unrealistic phenology routine. Clearly, optimization of the HSGSI parameters are needed in the future using data from multiple sites. In order to explain this phenomenon briefly, we inserted a new paragraph into the revised MS (page 30 lines 9-17 in the revised manuscript):
"It is notable that BBGCMuSo estimates too early onset of the vegetation growth, though the measured LAI is not available for the studied years. Taking into account the climate of the Bugac site, the intensive spring growth is expected to start around the beginning of March (Nagy et al., 2007; 2011) which is not reproduced by the BBGCMuSo simulations presented in Fig. 6d (for the original Biome-BGC prescribed, fixed start dates are used due to well recognized problems with the original grass phenology routine; see Hidy et al., 2012). BBGCMuSo uses the HSGSI method (see section 4.3.1) to estimate onset and offset of the growing season. In the presented simulation HSGSI parameter settings (most importantly the heatsum limit) were not adjusted, which resulted in too early onset due to warmer periods in the beginning of 2009 and 2010. Optimization of the HSGSI driving parameters might resolve this issue."

*It is well known that LAI tends to saturate earlier than net carbon uptake (e.g., abgC), which is well illustrated by observed values in the Mead1 site (Figure 7d, e). It seems that MuSo is capable of reproducing this pattern, both LAI and agbC shows a similar seasonal variability with peaks in a similar time period.*

Thank you for pointing to this issue. We agree; LAI and carbon uptake might decouple in ecosystems. We inserted a new paragraph into the text to discuss this problem (page 31 lines 23-26 in the revised manuscript):
"The measurement results from the Mead1 site show that abgC and LAI are decoupled after the initial growth, as LAI decreases but abgC still increases after the beginning of July. This means that LAI saturates earlier than biomass growth, but this pattern is not reflected by the models results, because abgC and LAI follows similar trend. This unrealistic behavior is likely the consequence of the static allocation parameterization."

*Because of these results, I'm skeptical about performance of the phenology model, particularly heat sum growing index (section 4.3.1). There is no need to modify model results, but instead I insist on describing these results as a limitation. They should be in a list for future modifications (P33 L18-22).*

According to the comment of the Referee we also extended the Discussion and Conclusion section with the following paragraph (page 34, lines 16-23 in the revised manuscript):
"Static allocation of C and N into the plant compartments is still present in BBGCMuSo, with a novel modification that uses GDD to initiate fruit allocation. Unrealistic coupling of aboveground biomass and LAI for the Mead simulation demonstrated the need to implement more flexible dynamic allocation. We plan to extend BBGCMuSo using the logic of the traditional crop models like DSSAT (Ritchie 1998) where GDD dependent allocation is implemented, and retranslocation of leaf C to fruit C is also realized during grain filling.

The Bugac case study demonstrated that the phenology module needs optimization to capture the observed onset of vegetation growth. This requires deeper evaluation of the HSGSI routine in multiple plant functional types. Extension of the HSGSI module might be also needed in the future."

**References**

[revised manuscript text omitted]

It is notable that BBGCMuSo estimates too early onset of the vegetation growth, though the measured LAI is not available for the studied years. Taking into account the climate of the Bugac site, the intensive spring growth is expected to start around the beginning of March (Nagy et al., 2007; 2011) which is not reproduced by the BBGCMuSo simulations presented in Fig. 6d (for the original Biome-BGC prescribed, fixed start dates are used due to well recognized problems with the original grass phenology routine; see Hidy et al., 2012). BBGCMuSo uses the HSGSI method (see section 4.3.1) to estimate onset and offset of the growing season. In the presented simulation HSGSI parameter settings (most importantly the heatsum limit) were not adjusted, which resulted in too early onset due to warmer periods in the beginning of 2009 and 2010. Optimization of the HSGSI driving parameters might resolve this issue.

[revised manuscript text omitted]

Static allocation of C and N into the plant compartments is still present in BBGCMuSo, with a novel modification that uses GDD to initiate fruit allocation. Unrealistic coupling of aboveground biomass and LAI for the Mead simulation demonstrated the need to implement more flexible dynamic allocation. We plan to extend BBGCMuSo using the logic of the traditional crop models like DSSAT (Ritchie 1998) where GDD dependent allocation is implemented, and retranslocation of leaf C to fruit C is also realized during grain filling.

The Bugac case study demonstrated that the phenology module needs optimization to capture the observed onset of vegetation growth. This requires deeper evaluation of the HSGSI routine in multiple plant functional types. Extension of the HSGSI module might be also needed in the future.

In the present paper three case studies were presented to demonstrate the effect of model structural improvements on the simulation quality. Clearly, extensive testing is required at multiple EC sites to evaluate the performance of the model and to make adjustments if needed.

Parameter uncertainty also needs further investigations. Model parameterization based on observed plant traits is possible today thanks to new datasets (e.g., White et al., 2000; Kattge et al., 2011; van Bodegom et al., 2012). However, in some cases, imperfect model structure may cause distortion owing to compensation of errors (Martre et al., 2015). Additionally, plant traits (e.g. specific leaf area, leaf C:N ratio, photosynthesis-related parameters) vary considerably in space so even though plant trait datasets are available they might not capture the site-level parameters. These issues raise the need for proper infrastructure for parameter estimation (or in other words, calibration or model inversion). The need for model calibration is also emphasized as we introduced a couple of empirical parameters in BBGCMuSo (e.g. those related to plant senescence, water stress thresholds and denitrification drivers, disturbance-related mortality parameters, phenology parameters) that need optimization using measurement data from multiple sites. In other words, though BBGCMuSo parameterization is mostly based on observable plant traits, model calibration can not be avoided in the majority of the cases (Hidy et al., 2012).

To address this problem we created a computer-based open infrastructure based on the workflow concept that can help a wide array of users in the application of the new model. The infrastructure is similar to PEcAn (Dietze et al., 2014) which is a collection of modules in a workflow that uses the Ecosystem Demography (ED v2.2) model. We followed the concept of model-data fusion (MDF; Williams et al., 2009) when developing the so-called "Biome-BGC Projects Database and Management System" (BBGCDB; http://ecos.okologia.mta.hu/bbgcdb/) and the BioVeL Portal (http://portal.biovel.eu) as a virtual research environment and collaborative tool. BBGCDB and the BioVeL portal help sensitivity analysis and parameter optimization of BBGCMuSo by a computer cluster-based Monte Carlo experiment and GLUE methodology (Beven and Binley, 1992). Further integration is planned using an alternative a Bayesian calibration algorithms (Hartig et al., 2012), that will provide extended calibration options to the parameters of BBGCMuSo.

We would like to support BBGCMuSo application for the wider scientific community as much as possible. For users of the original Biome-BGC, the question is of course the following: should I move to BBGCMuSo? Does it need a long learning process to move to the improved model? We would like to stress that users of the original Biome-BGC will have a smooth transition to BBGCMuSo, as we mainly preserved the structure of the input files. The majority of changes are related to the 7-layer soil module and to the implementation of management modules. The ecophysiological parameter file is similar to the original but it was extended considerably. This may raise issues but as it is emphasized in our User's Guide (Hidy et al., 2015), we extended the ecophysiological parameter file with many parameters that will not need adjustment in the majority of the cases. We simply moved some 'burned-in' parameters into the ecophysiological parameter in order to allow easy adjustment if needed in the future.

**Code availability**

The source code, the Windows model executable, sample simulation input files and documentation are available at the BBGCMuSo website (http://nimbus.elte.hu/bbgc). The source code is also available at GitHub (https://github.com/bpbond/Biome-BGC/tree/Biome-BGCMuSo_v4.0).

**Authors' Contributions**

Hidy developed Biome-BGCMuSo with modifying Biome-BGC 4.1.1 MPI version. The study was conceived and designed by Hidy and Barcza, with assistance from Marjanović, Churkina, Fodor, Horváth, and Running. It was directed by Barcza and Hidy. Marjanović, Ostrogović Sever, Pintér, Nagy, Gelybó, Dobor and Suyker contributed with simulations, measurement data and its post-processing and ideas on interpretation of the model validation. Hidy, Barcza and Marjanović prepared the manuscript and the supplement with contributions from all co-authors. All authors reviewed and approved the present article and the supplement. We are grateful for the Referees for their constructive comments that helped us to improve the quality of the manuscript.

[revised manuscript text omitted]